# LGG-1/GABARAP lipidation is not required for autophagy and development in *Caenorhabditis elegans*

**Romane Leboutet[1,2], Céline Largeau[1,2], Leonie Müller[3], Magali Prigent[1,2], Grégoire Quinet[4], Manuel S Rodriguez[4], Marie-Hélène Cuif[1,2], Thorsten Hoppe[3,5], Emmanuel Culetto[1,2], Christophe Lefebvre[1,2], Renaud Legouis[1,2]\***

[1]Institute for Integrative Biology of the Cell (I2BC), Université Paris-Saclay, CEA, CNRS, Gif-sur-Yvette, France; [2]INSERM U1280, Gif-sur-Yvette, France; [3]Institute for Genetics and Cologne Excellence Cluster on Cellular Stress Responses in Aging-Associated Diseases (CECAD), University of Cologne, Cologne, Germany; [4]Laboratoire de Chimie de Coordination (LCC), CNRS, Toulouse, France; [5]Center for Molecular Medicine Cologne (CMMC), Faculty of Medicine and University Hospital of Cologne, Cologne, Germany

**Abstract** The ubiquitin-like proteins Atg8/LC3/GABARAP are required for multiple steps of autophagy, such as initiation, cargo recognition and engulfment, vesicle closure and degradation. Most of LC3/GABARAP functions are considered dependent on their post-translational modifications and their association with the autophagosome membrane through a conjugation to a lipid, the phosphatidyl-ethanolamine. Contrarily to mammals, *C. elegans* possesses single homologs of LC3 and GABARAP families, named LGG-2 and LGG-1. Using site-directed mutagenesis, we inhibited the conjugation of LGG-1 to the autophagosome membrane and generated mutants that express only cytosolic forms, either the precursor or the cleaved protein. LGG-1 is an essential gene for autophagy and development in *C. elegans*, but we discovered that its functions could be fully achieved independently of its localization to the membrane. This study reveals an essential role for the cleaved form of LGG-1 in autophagy but also in an autophagy-independent embryonic function. Our data question the use of lipidated GABARAP/LC3 as the main marker of autophagic flux and highlight the high plasticity of autophagy.

**\*For correspondence:** renaud.legouis@i2bc.paris-saclay.fr

**Competing interest:** The authors declare that no competing interests exist.

## Editor's evaluation

The ubiquitin-like ATG8 family members act at multiple steps of autophagy, such as in autophagosome formation, cargo recognition and autophagosome maturation. ATG8 family members are lipidated that is thought to be required for their function. In this study, the authors provide evidence to show that the *C. elegans* ATG8 homolog LGG-1 possesses lipidation-independent function in autophagy, providing a novel insight into the role of ATG family members during animal development.

## Introduction

Macroautophagy is a highly dynamic vesicular degradation system that sequesters intracellular components in double membrane autophagosomes and delivers them to the lysosome (*Klionsky et al., 2021*). Upon induction, the successive recruitment of protein complexes triggers the phosphorylation of lipids, the transfer of lipids from various reservoirs, the recognition of cargoes, the tethering and the fusion (*Galluzzi et al., 2017*; *Nakatogawa, 2020*). One of the key players is the

ubiquitin-like protein Atg8, which in yeast is required for several steps during autophagy, such as initiation, cargo recognition and engulfment, and vesicle closure (*Kirisako et al., 2000*; *Knorr et al., 2014*; *Kraft et al., 2012*; *Nakatogawa et al., 2007*; *Xie et al., 2008*). There are seven isoforms of Atg8 homologs in humans defining two families, the MAP-LC3 (abbreviated as LC3A-a, LC3A-b, LC3B, LC3C) and the GABARAP (GABARAP, GABARAPL1, GABARAPL2; *Shpilka et al., 2011*). LC3/GABARAP proteins could have both similar and very specific functions during the autophagic flux (*Alemu et al., 2012*; *Grunwald et al., 2020*; *Joachim et al., 2015*; *Lystad et al., 2014*; *Pankiv et al., 2007*; *Weidberg et al., 2010*). LC3/GABARAP proteins can bind numerous proteins through specific motifs (LIR, LC3 interacting Region) and their interactomes are only partially overlapping (*Behrends et al., 2010*).

The pleiotropy of Atg8/LC3/GABARAP proteins in multiple cellular processes (*Galluzzi and Green, 2019*; *Schaaf et al., 2016*) entangles the study of their specific functions in human (*Nguyen et al., 2016*). Moreover, a series of post-translational modifications, similar to the ubiquitin conjugation, is involved in the membrane targeting of Atg8/LC3/GABARAP proteins. These proteins are initially synthesized as a precursor (P), then cleaved at their C-terminus after the invariant Glycine 116 (form I), and eventually conjugated to phosphatidylethanolamine (form II) at the membrane of autophagosomes (*Figure 1A*; *Kabeya et al., 2004*; *Kabeya et al., 2000*; *Scherz-Shouval et al., 2003*). Structural analyses have shown that LC3 /GABARAP can adopt an open or close configuration (*Coyle et al., 2002*). In addition, several other post-translational modifications have been reported, like phosphorylation (*Cherra et al., 2010*; *Herhaus et al., 2020*; *Wilkinson et al., 2015*), deacetylation (*Huang et al., 2015*) ubiquitination (*Joachim et al., 2017*) and oligomerization (*Chen et al., 2007*; *Coyle et al., 2002*), whose effects on LC3/GABARAP function and localization are largely unknown. The subcellular localization of Atg8/LC3/GABARAP proteins is either diffuse in the cytosol and nucleus, or associated to the membrane of various compartments or the cytoskeleton (*Schaaf et al., 2016*).

Due to such a versatile and pleiotropic repertoire, it is of particular interest to address the level of redundancy and specificity, including tissue specificity, of the various LC3/GABARAP members, and the possible functions of the forms P and I. In the nematode *Caenorhabditis elegans*, the presence of single homologs of LC3 and GABARAP, called respectively LGG-2 and LGG-1, represents an ideal situation to characterize their multiple functions (*Chen et al., 2017*; *Leboutet et al., 2020*).

The structure of LGG-1/GABARAP and LGG-2/LC3 is highly conserved (*Wu et al., 2015*) and both proteins are involved in autophagy processes during development, longevity, and stress (*Alberti et al., 2010*; *Chang et al., 2017*; *Chen et al., 2021*; *Meléndez et al., 2003*; *Samokhvalov et al., 2008*). In particular, the elimination of paternal mitochondria upon fertilization, also called allophagy (*Al Rawi et al., 2011*; *Sato and Sato, 2011*), has become a paradigm for dissecting the molecular mechanisms of selective autophagy (*Djeddi et al., 2015*; *Zhou et al., 2016*). Genetic analyses indicated that LGG-1 and LGG-2 do not have similar functions in autophagy (*Alberti et al., 2010*; *Jenzer et al., 2019*; *Manil-Ségalen et al., 2014*; *Wu et al., 2015*). During allophagy, LGG-1 is important for the recognition of ubiquitinated cargoes through interaction with the specific receptor ALLO-1 (*Sato et al., 2018*) and the formation of autophagosomes, whereas LGG-2 is involved in their maturation into autolysosomes and trafficking (*Djeddi et al., 2015*; *Manil-Ségalen et al., 2014*). LGG-1 and LGG-2 are also differentially involved during physiological aggrephagy in embryo, with temporal-specific and cargo-specific functions (*Wu et al., 2015*). Based on the presence of LGG-1 and LGG-2, three populations of autophagosomes have been described in *C. elegans* embryo: the major part are LGG-1 only, but LGG-2 only and double positives autophagosomes are also present (*Manil-Ségalen et al., 2014*; *Wu et al., 2015*). Moreover, LGG-1 is essential for embryonic and larval development, while LGG-2 is dispensable.

Using CRISPR-Cas9 editing, we investigated the functions of the non-lipidated cytosolic forms of LGG-1/GABARAP for bulk autophagy, mitophagy and aggrephagy, but also during starvation and longevity as well as apoptotic cell engulfment and morphogenesis. Here, we demonstrate that the non-lipidated form (LGG-1 I), but not the precursor form (LGG-1 P), is sufficient to maintain LGG-1 functions during development and aging. The cleavage of LGG-1 into form I is essential for autophagosome initiation and biogenesis while form II is involved in cargo recognition and autophagosome degradation.

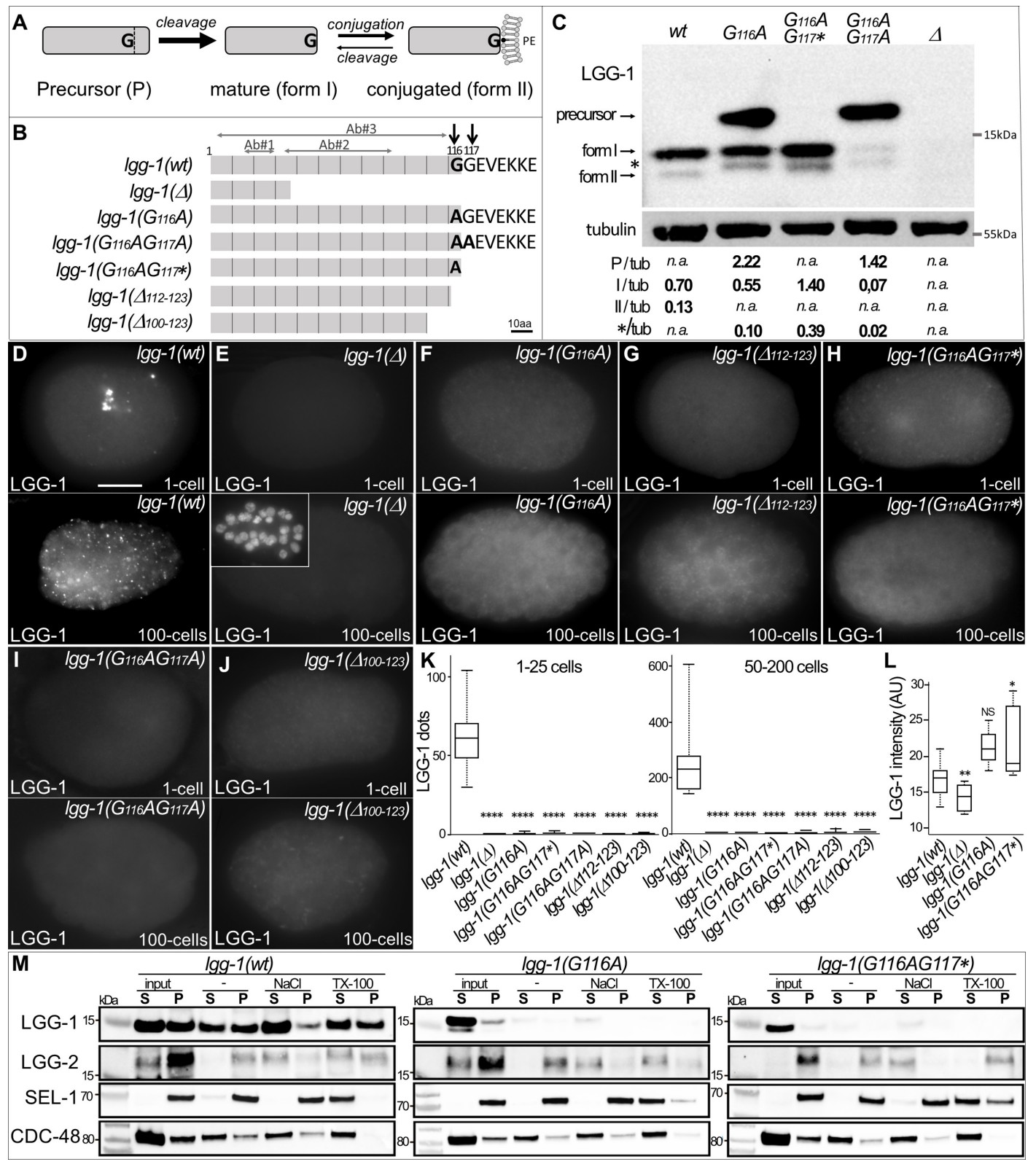

**Figure 1.** G116A abolishes the conjugation of LGG-1 to the membrane but not its cleavage. (**A**) Schematic representation of the various isoforms of Atg8s proteins after cleavage of the precursor and reversible conjugation to a phosphatidylethanolamine (PE). (**B**) Diagram of the theoretical proteins produced by the allelic *lgg-1* series used in this study. LGG-1(Δ) protein corresponds to the reference allele *lgg-1(tm3489)*, considered as a null, all others mutants have been generated using CRISP-Cas9. Black arrows point to the di-glycine residues which are mutated in alanine or stop codon (*).

*Figure 1 continued on next page*

*Figure 1 continued*

Other deletion mutants of the C-terminus result from non-homologous end joining. The mapping of the epitopes recognized by the LGG-1 antibodies (Ab#1, 2, 3) used in this study are indicated by horizontal grey arrows. (**C**) Western blot analysis of endogenous LGG-1 from total protein extracts of *wild-type, lgg-1(G116A), lgg-1(G116AG117*), lgg-1(G116AG117A), lgg-1(Δ)* young adults. The data shown is representative of three experiments using Ab#3 and was confirmed with Ab#1. The theoretic molecular mass of the precursor, and the form I are 14.8 kDa and 14.0 kDa, respectively, while the lipidated form II migrates faster. The asterisk indicates an unknown band. The quantification of each LGG-1 isoforms was normalized using tubulin. (**D–L**) Immunofluorescence analysis of endogenous LGG-1 (Ab#1 or Ab#2) in early and late embryos in *wild-type* (**D**), *lgg-1(Δ)* (**E**), *lgg-1(G116A)* (**F**), *lgg-1(Δ112–123)* (**G**), *lgg-1(G116AG117*)* (**H**), *lgg-1(G116AG117A)* (**I**), *lgg-1(Δ100–123)* (**J**). Inset in E shows the corresponding DAPI staining of nuclei. Box-plots quantification showing the absence of puncta in all *lgg-1* mutants (K, left n=19, 13, 11, 10, 6, 7, 6; right n=18, 14, 12, 10, 10, 9, 12) and the increase of cytosolic staining in *lgg-1(G116A)* and *lgg-1(G116AG117*)* (L, n=19, 13, 11, 10). Kruskal Wallis test, p-value *<0.05, **<0.01, ****<0.0001, NS non-significant. Scale bar is 10 μm. (**M**) Cellular fractionation of membrane vesicles. Western blot analysis for detection of LGG-1 together with LGG-2 (autophagosome marker), SEL-1 (ER marker), and CDC-48 (ER-associated and cytosol) using supernatant (**S**) and pellet (**P**) fractions of *lgg-1 wild-type, lgg-1(G116A)*, and *lgg-1(G116AG117*)* worm lysates treated with fractionation buffer (-), sodium chloride (NaCl) or Triton X-100 (TX-100) after subcellular fractionation. Proteins associated with membranes are solubilized by NaCl, and resident proteins in membrane-bound organelles are released only by dissolving the membrane with detergents. While wild-type LGG-1 is detected in the cytosolic fraction (input S) and in the various membrane fractions, mutant LGG-1 is almost exclusively present in the cytosolic fraction in *lgg-1(G116A)* and *lgg-1(G116AG117*)*.

The online version of this article includes the following source data and figure supplement(s) for figure 1:

**Source data 1.** Folder containing original microscopy pictures, quantification data and western blots shown in *Figure 1*.

**Figure supplement 1.** Description of *lgg-1* alleles.

**Figure supplement 2.** Identification of LGG-1(G116A) and LGG-1(G116AG117*) forms.

**Figure supplement 2—source data 1.** Folder containing original microscopy pictures and quantification data shown in *Figure 1—figure supplement 2*.

**Figure supplement 3.** GFP::LGG-1(G116A) does not localize to autophagosomes.

**Figure supplement 3—source data 1.** Folder containing original microscopy pictures and quantification data shown in *Figure 1—figure supplement 3*.

## Results

### The G116G117 di-Glycine motif is a substrate for cleavage of LGG-1 precursor

The LGG-1 protein is highly conserved from residue 1 to residue 116, sharing 92% and 74% similarity with the human GABARAP and the yeast Atg8, respectively (*Manil-Ségalen et al., 2014*). However, the GEVEKKE C-terminus of LGG-1 is unusual by its length and the presence of a non-conserved glycine residue in position 117 (*Figure 1B*, *Figure 1—figure supplement 1*). As consistent with other *Caenorhabditis* species as well as several nematodes and arthropods, the presence of a C-terminal di-glycine is reminiscent of other ubiquitin-like proteins such as SUMO and Nedd8 (*Cappadocia and Lima, 2018*; *Jentsch and Pyrowolakis, 2000*). These specificities raise the possibility that the C-terminus could confer particular functions to the precursor and the cleaved form.

To analyze the functions of LGG-1 P and LGG-1 I, a CRISPR-Cas9 approach was used to substitute the conserved glycine 116 by an alanine, and to generate three specific *lgg-1* mutants with various C-terminus (*Figure 1B*). In theory, both *lgg-1(G116A)* and *lgg-1(G116AG117A)* mutants were expected to accumulate a P form due to the blockage of its cleavage by ATG-4 (*Wu et al., 2012*). Alternatively, the *lgg-1(G116AG117*)* mutant should produce a form I. Five supplementary *lgg-1* frameshift mutants were isolated during the CRISPR experiments, resulting in deletion/insertion at the C-terminus (*Figure 1B* and *Figure 1—figure supplement 1*). Among them, *lgg-1(ΔC100-123)* and *lgg-1(ΔC112-123)* have been used in the present study. The allele *lgg-1(tm3489)*, which deletes 48% of the open reading frame, was used as a negative control (*Manil-Ségalen et al., 2014*) as it is considered as a null mutant, and thereafter noted *lgg-1(Δ)*.

To assess whether *lgg-1(G116A)*, *lgg-1(G116AG117A)* and *lgg-1(G116AG117*)* alleles code for a precursor and form I, respectively, we performed a western blot analysis with two different LGG-1 antibodies (*Al Rawi et al., 2011*; *Springhorn and Hoppe, 2019*; *Figure 1C*). In basal conditions the wild-type LGG-1 was mainly present as form I (13.9 kDa) with a low amount of the faster migrating form II and no detectable precursor (14,8 kDa)(*Figure 1C*), while no band was observed in the allele *lgg-1(tm3489)* confirming that it is a *bona fide* null mutant. While the *lgg-1(G116AG117*)* mutant presented a major form I, the *lgg-1(G116A)* mutant accumulated both the expected precursor form

and form I. This indicated that the cleavage of the LGG-1(G116A) precursor was still present although less efficient. In both mutants, an unexpected minor form was observed migrating differently from the lipidated form II, which was no longer detected. The *lgg-1(G116AG117A)* mutant accumulated the precursor form (96% of the protein) indicating that the cleavage observed in the LGG-1(G116A) was dependent on the presence of a second glycine in position 117.

The respective protein substitutions were further confirmed by mass spectrometry analyses after affinity purification of LGG-1(G116A) and LGG-1(G116AG117*) (*Figure 1—figure supplement 2*). The identification of C-terminal peptides validated the expected precursor form in LGG-1(G116A) and its cleavage after A116, and confirmed A116 as the last residue in LGG-1(G116AG117*). These latter forms are called hereafter 'cleaved form' and 'truncated form', respectively.

## Glycine 116 is essential for lipidation of LGG-1 after cleavage

To confirm western blot analyses, we next performed immunofluorescence in the embryo to analyze the subcellular localization of LGG-1 protein from the various alleles. At the one-cell-stage and around 100 cells-stage, two selective autophagy processes have been well characterized, removing paternal mitochondria and maternal aggregates, respectively (*Al Rawi et al., 2011*; *Sato and Sato, 2011*; *Zhang et al., 2009*). The punctate staining, observed in the wild-type animals (*Figure 1D*) with two independent anti-LGG-1 antibodies, was characteristic for the autophagosomes formed during each process, and was absent in the *lgg-1(Δ)* mutant (*Figure 1E*). The five mutants *lgg-1(G116A)*, *lgg-1(G116AG117*)*, *lgg-1(G116AG117A)*, *lgg-1(ΔC100-123)*, and *lgg-1(ΔC112-123)* presented no puncta but a diffuse cytosolic staining. (*Figure 1F–K*), indicating that neither the precursor nor the form I are able to conjugate to the autophagosome membrane. The increase of the diffuse signal in *lgg-1(G116A)* and *lgg-1(G116AG117*)* embryos (*Figure 1L*) suggests that the protein is less degraded in these mutants. Moreover, no LGG-1(G116A) puncta were observed after depleting the tethering factor EPG-5 compared to the strong accumulation of puncta in LGG-1(wt) (*Figure 1—figure supplement 2*; *Tian et al., 2010*).

We performed cellular fractionation of membrane vesicles to test whether LGG-1(G116A) and LGG-1(G116AG117*) are associated with autophagosomes. Compared with ER resident SEL-1 or ER-associated CDC-48, the LGG-1(wt) protein was detected in both the cytosolic fraction and the membrane pellet and could only be extracted with high salt or the detergent Triton X-100. In contrast to LGG-1(wt), both LGG-1(G116A) and LGG-1(G116AG117*) were absent in the membrane pellet fraction (*Figure 1M*), suggesting defective lipidation of both LGG-1 mutant proteins. In an alternative approach, we observed the localization of overexpressed GFP::LGG-1 and GFP::LGG-1(G116A) (*Manil-Ségalen et al., 2014*) after induction of autophagic flux by acute heat stress (aHS) (*Chen et al., 2021*; *Kumsta et al., 2017*). After aHS, GFP::LGG-1 formed numerous puncta that further accumulated when autolysosome formation was impaired by depletion of RAB-7 or EPG-5 (*Figure 1—figure supplement 3*). In contrast, in GFP::LGG-1(G116A), puncta were not reduced under any condition. Electron microscopy and immunogold labeling confirmed that GFP::LGG-1 was frequently detected to autophagosome membranes (*Manil-Ségalen et al., 2014*), whereas GFP::LGG-1(G116A) was rarely detected in association with autophagosomes and in these rare cases was predominantly localized in the lumen (*Figure 1—figure supplement 3*). Taken together, these results suggest that the G116A mutation does not allow conjugation of LGG-1 to the autophagosome membrane despite its cleavage. LGG-1(G116AG117A) represents only a precursor form and LGG-1(G116AG117*) only a truncated form, whereas LGG-1(G116A) produces both a precursor and a cleaved form. This allele series provides an ideal situation to study the respective roles of the precursor and form I in absence of lipidated form II.

## The essential function of LGG-1 during development is dependent of its cleavage but not its conjugation

The developmental phenotypes of the mutants *lgg-1(G116A)*, *lgg-1(G116AG117A)*, and *lgg-1(G116AG117*)* were explored in embryo, larvae, and adults and compared with *lgg-1(Δ)* and wild-type animals (*Figure 2*). We confirmed that *lgg-1(Δ)* homozygous animals present a massive lethality during late embryogenesis or first larval stage (*Figure 2B and H*; *Manil-Ségalen et al., 2014*). However, few escapers, circa 8% of the progeny, were able to reach adulthood and reproduce, allowing to maintain a *lgg-1(Δ)* homozygous population.

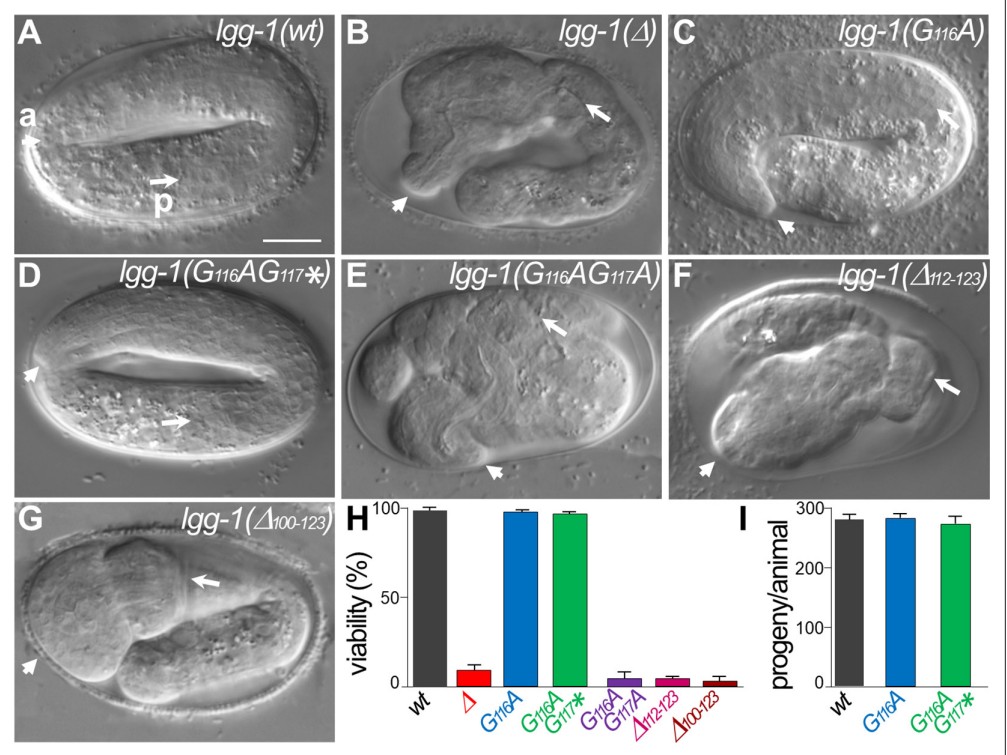

**Figure 2.** *lgg-1(G116A)* and *lgg-1(G116AG117*)* mutants are viable with no developmental defect. (**A–G**) DIC images of embryos after morphogenesis in *wild-type* (**A**), *lgg-1(Δ)* (**B**), *lgg-1(G116A)* (**C**), *lgg-1(G116AG117*)* (**D**), *lgg-1(G116AG117A)* (**E**), *lgg-1(Δ112–123)* (**F**), *lgg-1(Δ100–123)* (**G**). *lgg-1(G116AG117A)*, *lgg-1(Δ112–123)*, *lgg-1(Δ100–123)*, *lgg-1(Δ)* mutant embryos present severe developmental defects. Short and long white arrows point to the anterior (**a**) and posterior (**p**) part of the pharynx, respectively. Scale bar is 10 μm. (**H**) The viability, expressed as the percentage of embryos reaching adulthood, is not affected in *lgg-1(G116A)* and *lgg-1(G116AG117*)* mutants (42<n < 103). (**I**) The fertility, total number of progenies, of *lgg-1(G116A)* and *lgg-1(G116AG117*)* adults is similar to *wild-type* (n=20).

The online version of this article includes the following source data for figure 2:

**Source data 1.** Folder containing original microscopy pictures and quantification data shown in **Figure 2**.

---

Neither *lgg-1(G116A)* nor *lgg-1(G116AG117*)* homozygous animals presented any observable defect in development (**Figure 2C, D and H**) or adulthood and they reproduced at a similar rate compared to wild-type animals (**Figure 2I**). In contrast, *lgg-1(G116AG117A)* and the five independent mutants harboring various deletions and frameshifts of the C-terminus presented a very strong lethality with the characteristic embryonic phenotype of *lgg-1(Δ)* animals (**Figure 2E–H**). Among them *lgg-1(Δ112–123)* presented a premature stop codon at position 112 and two others a frameshift in position 114 leading to an extension of the C-terminus (**Figure 2** and **Figure 1—figure supplement 1**).

These data indicate that the cleaved LGG-1(G116A) and the truncated LGG-1(G116AG117*) forms, but not the precursor, are sufficient to recapitulate the normal development and viability, independently of membrane conjugation. These data suggest that cleavage of the C-terminus is necessary for LGG-1 developmental functions.

## Autophagy is functional in LGG-1(G116A)

To address the functionality of LGG-1 precursor and form I, we analyzed autophagy-related processes that have been well characterized during *C. elegans* life cycle (**Leboutet et al., 2020**). Selective autophagy was studied in the early embryo, where a stereotyped mitophagy process occurs. The degradation of selective cargos was observed in live embryos using specific labeling of the paternal mitochondria (HSP-6::GFP and mitoTracker, **Figure 3A–F** and **Figure 3—figure supplement 1**). In *lgg-1(Δ)* animals, the cargos accumulated while they were degraded in the wild-type situation.

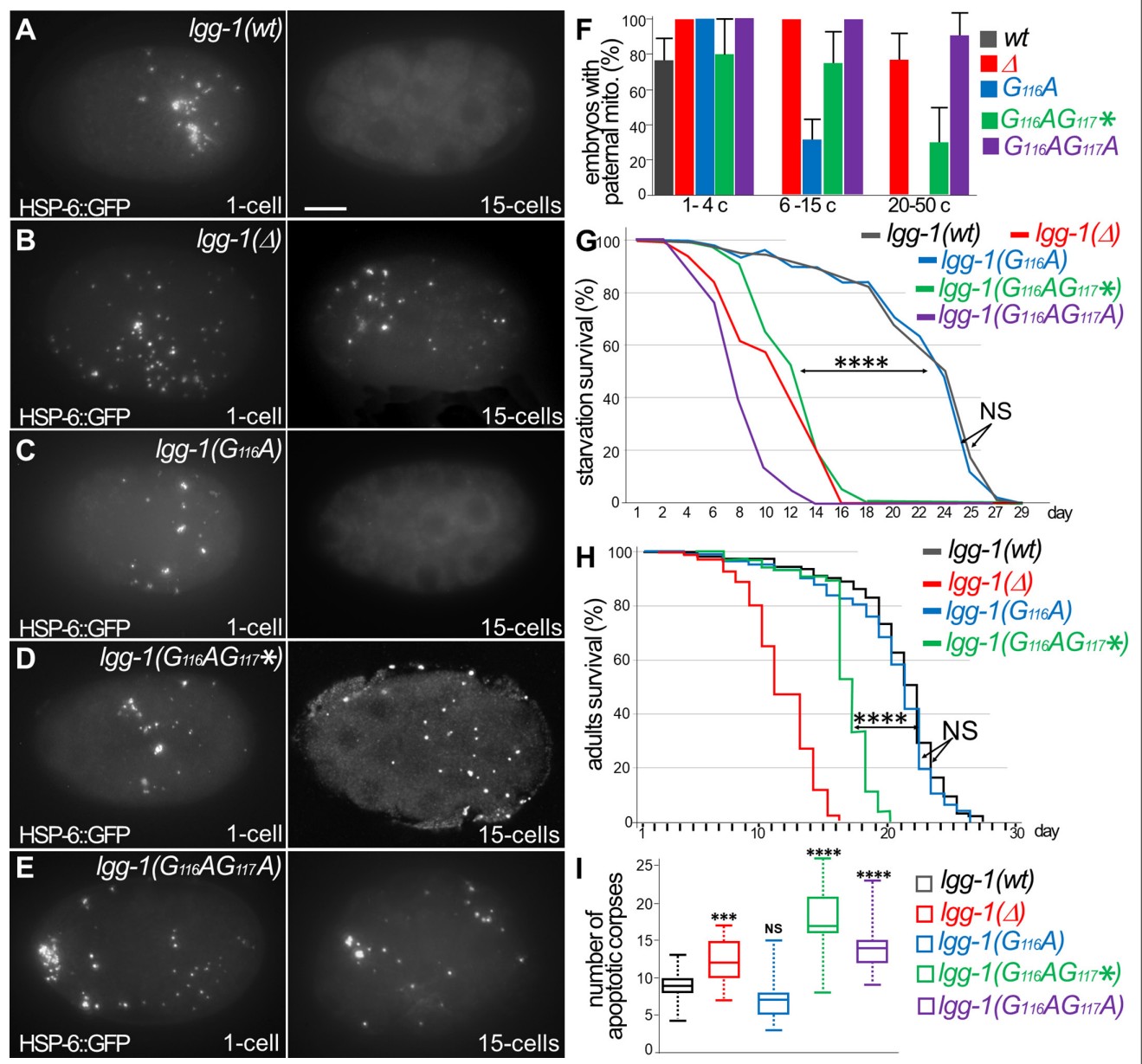

**Figure 3.** Autophagy is functional in *lgg-1(G116A)* but not in *lgg-1(G116AG117\*)* and l*gg-1(G116AG117A)*. (**A–E**) In vivo epifluorescence images of paternal mitochondria (HSP-6::GFP) at the 1 cell and 15 cells stages in *wild-type* (**A**), *lgg-1(Δ)* (**B**), *lgg-1(G116A)* (**C**), *lgg-1(G116AG117\*)*(**D**), *lgg-1(G116AG117A)*(**E**) embryos showing an effective degradation of paternal mitochondria in *wt* and *lgg-1(G116A)* but not in *lgg-1(Δ) lgg-1(G116AG117\*)* and *lgg-1(G116AG117A)*. Quantification are shown in (**F**). (**G, H**) Bulk autophagy during aging and stress was assessed by worm longevity (G, log rank test n>100 animals, p-value ****<0.001) and starvation survival (H, Chi-square test at day 15 p-value ****<0.001. The survival is significantly reduced in *lgg-1(Δ)*, *lgg-1(G116AG117\*)* and *lgg-1(G116AG117A)* compared to *wt* and *lgg-1(G116A)*. NS non-significant. (**I**) Box-plots quantification of apoptotic corpses showing a defective degradation in *lgg-1(G116AG117\*)* and *lgg-1(G116AG117A)* but not in *lgg-1(G116A)* (n=22, 40, 46, 14, 21 Kruskal Wallis test ***<0.001, ****<0.0001, NS non-significant).

The online version of this article includes the following source data and figure supplement(s) for figure 3:

**Source data 1.** Folder containing original microscopy pictures and quantification data shown in *Figure 3*.

**Figure supplement 1.** Autophagy is functional in *lgg-1(G116A)* but not in *lgg-1(G116AG117\*)*.

**Figure supplement 1—source data 1.** Folder containing original microscopy pictures and quantification data shown in *Figure 3—figure supplement 1*.

**Figure supplement 2.** Atg8(G116A) and Atg8(G116AR117\*) are functional for vacuolar shaping but not for autophagy in *S. cerevisiae*.

**Figure supplement 2—source data 1.** Folder containing original microscopy pictures and quantification data shown in *Figure 3—figure supplement 2*.

In *lgg-1(G116A)*, but neither in *lgg-1(G116AG117\*)* nor in *lgg-1(G116AG117A)* mutants, paternal mitochondria were degraded, suggesting that the LGG-1(G116A) protein maintained autophagic activity.

Bulk autophagy was then studied by starvation of the first stage larvae (**Figure 3G**). While *lgg-1(G116AG117\*)* and *lgg-1(G116AG117A)* mutants displayed a marked decrease of survival, *lgg-1(G116A)* mutants showed no difference compared the wild-type animals. Moreover, the longevity of adults, which depends on bulk autophagy, was similar for *lgg-1(G116A)* and wild-type animals (**Figure 3H**), but strongly reduced for *lgg-1(G116AG117\*)* mutants.

The autophagic capacity of LGG-1(G116A) protein, but not LGG-1(G116AG117\*) or LGG-1(G116AG117A), was further documented by the elimination of apoptotic corpses in the embryo (**Figure 3I**, and **Figure 3—figure supplement 1**; *Jenzer et al., 2019*).

Overall, these data demonstrate that, despite its defect to localize to autophagosomes, LGG-1(G116A) achieves both selective and bulk autophagy during physiological and stress conditions. This is the first in vivo evidence that the autophagy functions of LGG-1/GABARAP can be uncoupled from its membrane conjugation. The non-functionality of LGG-1(G116AG117A) suggests that the precursor form is not responsible of LGG-1(G116A) autophagy activity. Despite an identical protein sequence, the truncated LGG-1(G116AG117\*) is not functional in autophagy, indicating that the cleavage of the C-terminus from the precursor is essential for the functionality of LGG-1(G116A). Moreover, the normal development of *lgg-1(G116AG117\*)* animals demonstrates that the developmental functions of LGG-1 are independent of its autophagic functions. Interestingly, the expression in *S. cerevisiae* of LGG-1(wt) and LGG-1(G116A), but not LGG-1(G116AG117\*), slightly improved the nitrogen starvation survival of *atg8Δ* mutant (Supplementary data and **Figure 3—figure supplement 2**), suggesting that the LGG-1(G116A) retains a partial autophagy functionality in the yeast.

## Autophagy but not developmental functions of LGG-1(G116A) partially depends on LGG-2

Our previous study has shown a partial redundancy of LGG-1 and LGG-2 during starvation survival, and longevity (*Alberti et al., 2010*), which raises the possibility of functional compensation of *lgg-1(G116A)* by LGG-2. To test this possibility, we used the large deletion mutant *lgg-2(tm5755)*, which is considered as a null (*Manil-Ségalen et al., 2014*), and constructed the double mutant strains *lgg-1(G116A); lgg-2(tm5755)* and *lgg-1(G116AG117\*); lgg-2(tm5755)*.

Similar to the single mutants *lgg-1(G116A)* and *lgg-2(tm5755)*, the double mutant *lgg-1(G116A); lgg-2(tm5755)* animals were viable and presented no morphological defect (**Figure 4A–F**). These data indicate that the correct development of *lgg-1(G116A)* is not due to a compensative mechanism involving *lgg-2*.

Next, we analyzed the autophagy functions in *lgg-1(G116A); lgg-2(tm5755)* animals. If LGG-2 compensates for LGG-1(G116A) in autophagy, *lgg-1(G116A); lgg-2(tm5755)* animals should behave similarly to *lgg-1(G116AG117\*); lgg-2(tm5755)* (of note *lgg-1(Δ); lgg-2(tm5755)* animals are not viable). The *lgg-1(G116A); lgg-2(tm5755)* animals presented a decrease for both survival to starvation and longevity compared to *lgg-1(G116A)* single mutant. However, they survived better than *lgg-1(G116AG117\*); lgg-2(tm5755)* animals (**Figure 4G and H**). These results indicate that the functionality of LGG-1(G116A) in bulk autophagy partially relies on LGG-2. Selective autophagy during early embryogenesis was then quantitatively analyzed in the double mutant strains (**Figure 4I–K**). Surprisingly, paternal mitochondria were eliminated in the *lgg-1(G116A); lgg-2(tm5755)* animals indicating that LGG-1(G116A) was sufficient for the allophagy process. This suggests that paternal mitochondria could be degraded by autophagosomes devoid of both LGG-1 and LGG-2. However, a delay in the degradation was observed compared to *lgg-1(G116A)* animals suggesting that the autophagy flux is reduced. These results revealed a partial redundancy between LGG-1 and LGG-2 in autophagy, but demonstrated at the same time that LGG-1(G116A) fulfills developmental functions and maintains some autophagy activity independent of LGG-2.

Interestingly, this detailed analysis also revealed a slight delay in the elimination of paternal mitochondria in *lgg-1(G116A)* animals compared to wild-type (**Figure 4K**). Although the cleaved LGG-1 is sufficient for autophagy, this observation suggests that loss of membrane targeting could affect the dynamics of autophagy flux.

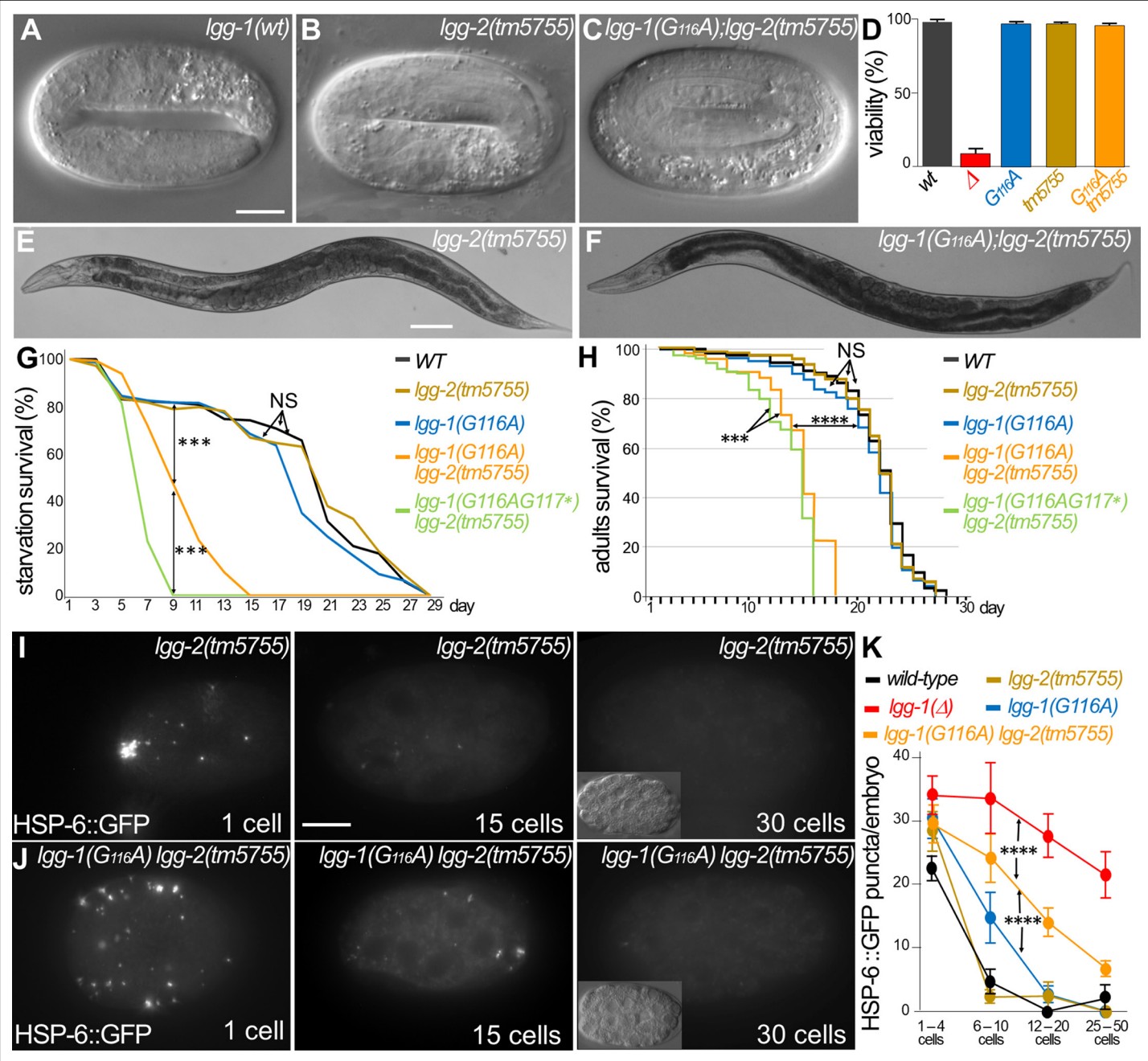

**Figure 4.** Autophagy but not developmental function of LGG-1(G116A) partially depends on LGG-2. (A–F) DIC images of embryos and bright field images of adults in *wild-type* (A), *lgg-2(tm5755)* (B, E), *lgg-1(G116A); lgg-2(tm5755)* (C, F). The double mutant *lgg-1(G116A); lgg-2(tm5755)* animals have no morphogenetic defects and no decrease in viability compare to single mutants or the *lgg-1(Δ)* (quantification in D). (G–H) Bulk autophagy during stress and aging was assessed by starvation survival (G, Chi-square test at day 9 ***p-value <0.001) and worm longevity (H, log rank test n>100 animals, ***p-value <0.001, ****p-value <0.0001). The survival of double mutants *lgg-1(G116A); lgg-2(tm5755)* and *lgg-1(G116AG117*); lgg-2(5755)* is reduced compared to *wild-type* and single mutant *lgg-1(G116A)* and *lgg-2(tm5755)*. *lgg-1(G116A); lgg-2(tm5755)* animals survive to starvation better than *lgg-1(G116AG117*); lgg-2(5755)* and present a slightly higher lifespan. (I–K) In vivo epifluorescence imaging of paternal mitochondria (HSP-6::GFP) at the 1 cell, 15 cells, and 30 cells stages in *lgg-2(tm5755)*, (I) *lgg-1(G116A); lgg-2(tm5755)* (J) embryos and quantification (n=50, 39, 35, 45, 46 Chi-square test ****<0.0001) (K). Elimination of mitochondria is efficient but delayed in *lgg-1(G116A); lgg-2(5755)* compared to *lgg-1(G116A)*. Insets show the corresponding DIC pictures. Scale bar is 10 µm (A–C, I, J) or 100 µm (E, F).

The online version of this article includes the following source data for figure 4:

**Source data 1.** Folder containing original microscopy pictures and quantification data shown in *Figure 4*.

## The degradation of autophagosomes is delayed in LGG-1(G116A)

The autophagic flux and the dynamics of autophagosome formation were compared between *lgg-1(G116A)*, *lgg-1(G116AG117\*)* and *lgg-1(Δ)* animals. We first focused on the early embryo where the autophagy process is stereotyped and the nature of the cargos and the timing of degradation are well characterized. Moreover, the autophagosomes sequestering the paternal mitochondria were clustered and positive for LGG-2 (*Figure 5A*; *Manil-Ségalen et al., 2014*). In *lgg-1(Δ)* mutant, LGG-2 autophagosomes were not detected as a cluster but were spread out in the whole embryo as single puncta that persisted after the 15 cells stage (*Figure 5B and E*). This indicated that individual LGG-2 structures could be formed in absence of LGG-1, but were not correctly localized and not degraded properly, presumably because of the role of LGG-1 in cargo recognition (*Sato et al., 2018*) and of its latter involvement in the maturation of autophagosomes, respectively. The pattern of LGG-2 was somehow different in *lgg-1(G116A)* and *lgg-1(G116AG117\*)* mutants, forming sparse structures of heterogeneous size, which persisted longer (*Figure 5C–F*). These data suggested that the cleaved and the truncated LGG-1 could both promote the recruitment of LGG-2 to autophagic structures, but display an altered autophagic flux. The analysis of the colocalization between paternal mitochondria and LGG-2 did not reveal an increase in *lgg-1(G116A)* or *lgg-1(G116AG117\*)* mutants (*Figure 5G–J* and *Figure 5—figure supplement 1*). These data suggested that the elimination of paternal mitochondria in *lgg-1(G116A)* animals was not due to the enhanced recruitment of LGG-2. A western blot analysis of worm lysates indicated that there was no increase of LGG-2 expression in *lgg-1(Δ)*, *lgg-1(G116A)*, and *lgg-1(G116AG117\*)* mutants (*Figure 5K*).

The autophagic structures in *lgg-1(G116A)* and *lgg-1(G116AG117\*)* embryos were further characterized by electron microscopy and compared with wild-type and *lgg-1(Δ)* mutant embryos (*Figure 6*). In wild-type animals, autophagosomes containing cytoplasmic materials (referred as type 1) and the characteristic paternal mitochondria (*Zhou et al., 2016*) were observed in early embryos (*Figure 6A*). At that stage, rare autophagosomes containing partially degraded material were present (referred as type 2). As expected, almost no autophagosome was observed in *lgg-1(Δ)* embryos and paternal mitochondria were non-sequestered (*Figure 6B*). In *lgg-1(G116A)* embryos, the numbers of type 1 and type 2 autophagosomal structures increased. The autophagosomes appeared to be closed and contained various cellular materials and membrane compartments (*Figure 6C–E*). This confirmed that LGG-1(G116A) was sufficient to form functional autophagosomes but with delayed degradation. On the other hand, *lgg-1(G116AG117\*)* embryos presented non-sequestered paternal mitochondria (*Figure 6F*) and multi-lamellar structures containing cytoplasm but no membrane organelles (type 3 *Figure 6G and K*). The analysis of the double mutant strains *lgg-1(G116A); lgg-2(tm5755)* and *lgg-1(G116AG117\*); lgg-2(tm5755)* revealed the presence of types 1 and 2 autophagosomes, but less frequent than in the single *lgg-1* mutants (*Figure 6H–K*). This data confirmed that LGG-1(G116A) alone was able to initiate the formation of autophagosomes but less efficiently in absence of LGG-2. Type 3 structures were only observed in *lgg-1(G116AG117\*)* and *lgg-1(G116AG117\*); lgg-2(tm5755)*, suggesting a neomorphic function of the truncated LGG-1(G116AG117\*) protein that induced a non-functional compartment.

Altogether, these data indicate that the cleaved, but not the truncated, LGG-1 form I is able to form functional autophagosomes with a delayed degradation.

## The lipidated LGG-1 is involved in the coordination between cargo recognition and autophagosome biogenesis

To better understand the function of LGG-1 form I during autophagy flux, we next analyzed a developmental aggrephagy process (*Figure 7*). The Zhang lab has demonstrated that aggregate-prone proteins are degraded through autophagy in *C. elegans* embryo through liquid-liquid phase separation promoted by the receptor SEPA-1 and regulated by the scaffolding protein EPG-2 (*Lu et al., 2011*; *Tian et al., 2010*; *Wu et al., 2015*; *Zhang et al., 2018*; *Zhang et al., 2009*). Initiation and elongation of autophagosomes were analyzed by quantifying the colocalization between ATG-18/WIPI2 and LGG-2 (*Figure 7A–E*) during autophagosome formation. ATG-18, the worm homolog of the omegasome marker WIPI2 (*Polson et al., 2010*), acts at an early step of biogenesis (*Lu et al., 2011*). Puncta labelled with ATG-18 only, both ATG-18 and LGG-2, or LGG-2 only were considered as omegasomes, phagophores, and autophagosomes, respectively. In *lgg-1(RNAi)* animals the number of omegasomes increased while the proportion of phagophore decreased compared to the wild-type

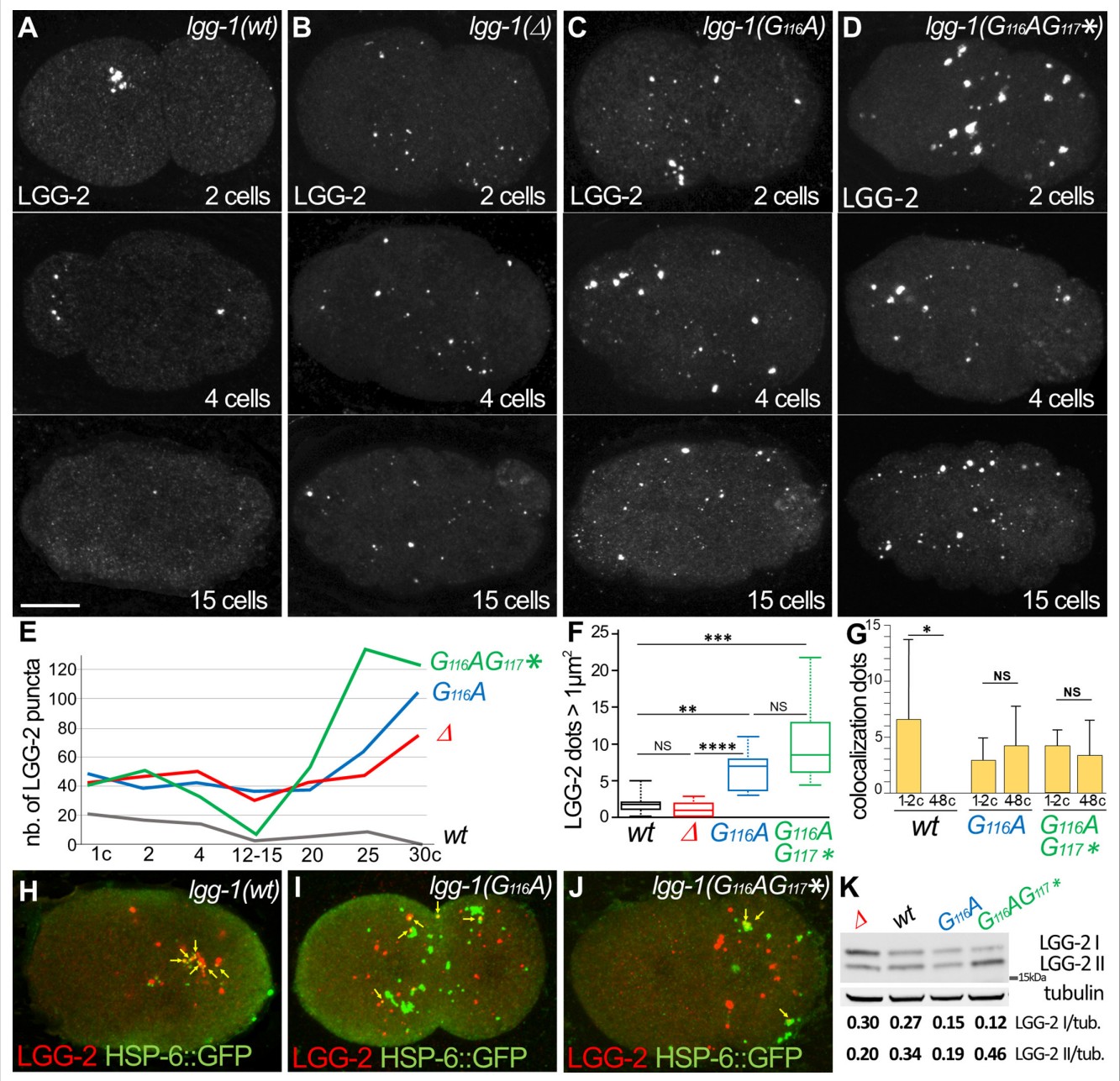

**Figure 5.** The degradation of autophagosomes is delayed in *lgg-1(G116A)*. (**A–F**) Confocal images of LGG-2 immunofluorescence in 2 cells, 4 cells, and 15 cells in *wild-type* (**A**), *lgg-1(Δ)* (**B**), *lgg-1(G116A)* (**C**), *lgg-1(G116AG117*)* (**D**) and quantification of the number (**E**) and size of puncta (**F**) (embryo analyzed 19, 37, 28, 14; Mann-Whitney test, p-value ****<0.0001). In *lgg-1(G116A)* and *lgg-1(G116AG117*)* mutants LGG-2 is detected as heterogeneous sparse structures that persist. (**G–J**) Colocalization analysis of paternal mitochondria (HSP-6::GFP) and LGG-2 puncta (**H**) from confocal images of *wild-type* (**H**), *lgg-1(G116A)* (**I**) and *lgg-1(G116AG117*)* (**J**) early embryos. (Mean + SD, n=16, 20, 12, Kruskal Wallis test p-value*<0.05). The clustering of paternal mitochondria and LGG-2 autophagosomes are absent in *lgg-1(G116A)* and *lgg-1(G116AG117*)* where HSP-6::GFP and LGG-2 puncta are mainly separated with rare colocalization events (yellow arrows). (**K**) Western blot analysis of endogenous LGG-2 from total protein extracts from *wild-type, lgg-1(G116A), lgg-1(G116AG117*), lgg-1(Δ)* young adults. The quantification of LGG-2 upper and lower bands was normalized using tubulin.

The online version of this article includes the following source data and figure supplement(s) for figure 5:

**Source data 1.** Folder containing original microscopy pictures, quantification data and western blots shown in *Figure 5*.

**Figure supplement 1.** Colocalization quantification of HSP-6::GFP and LGG-2.

**Figure supplement 1—source data 1.** Folder containing quantification data shown in *Figure 5—figure supplement 1*.

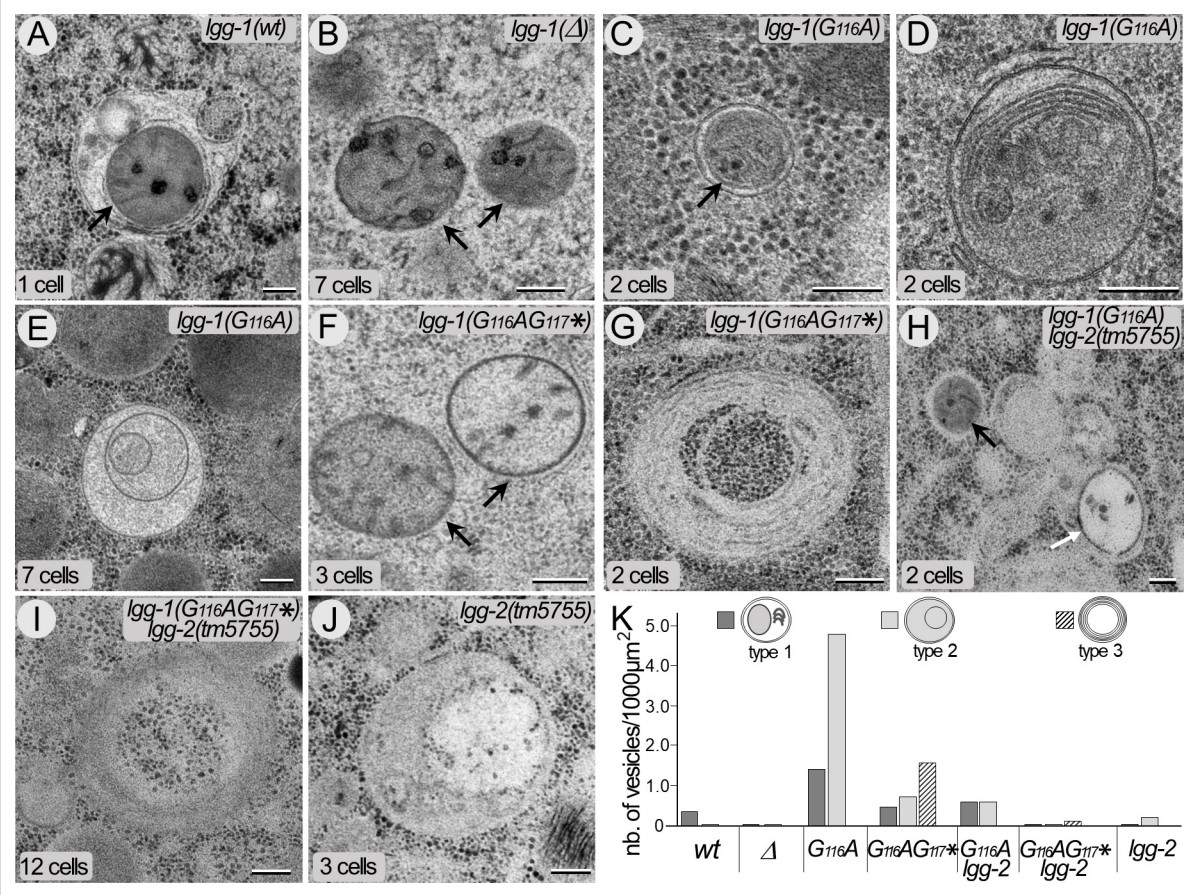

**Figure 6.** The cleaved LGG-1 is sufficient for autophagosome biogenesis. (**A–J**) Electron microscopy images of autophagosomes in *wild-type* (**A**), *lgg-1(Δ)* (**B**), *lgg-1(G116A)* (**C–E**), *lgg-1(G116AG117\*)* (**F–G**), *lgg-1(G116A); lgg-2(tm5755)* (**H**), *lgg-1(G116AG117\*) lgg-2(tm5755)* (**I**) and *lgg-2(tm5755)* (**J**) early embryos. Type 1 autophagosomes (**A, C, D**) appear as closed structures containing various membrane organelles. Among those, sequestered paternal mitochondria (black arrows) are observed in *wild-type* and *lgg-1(G116A)* embryos but remain unsequestered in *lgg-1(Δ)* and *lgg-1(G116AG117\*)* embryos. Type 2 autophagosomes (**E**, white arrow in **H, J**) appear as closed structures containing unidentified or degraded materials. Type 3 structures (**G, I**) are multi-lamellar structures only detected in *lgg-1(G116AG117\*)* embryos. Scale bar is 200 nm. (**K**) Quantification of type 1, type 2, and type 3 structures in early embryo (1–12 cells). In *lgg-1(G116A)* embryos, the numbers of type 1 and type 2 autophagosomal structures increase supporting a retarded degradation. The formation of autophagosomes in *lgg-1(G116A)* and *lgg-1(G116AG117\*)* embryos is partially dependent of LGG-2 (n sections = 32, 62, 32, 19, 32, 26, 52).

The online version of this article includes the following source data for figure 6:

**Source data 1.** Folder containing original microscopy pictures and quantification data shown in *Figure 6*.

embryos (*Figure 7A, B and E*). This indicates that the initiation of autophagy was triggered in absence of LGG-1, but the biogenesis of autophagosome was defective. *lgg-1(G116A)* animals showed no difference with the wild-type (*Figure 7C and E*) supporting that both initiation and phagophore extension are normal with the cleaved LGG-1. Similar to the *lgg-1(RNAi)*, *lgg-1(G116AG117\*)* animals were defective in the phagophore extension (*Figure 7D and E*). These data confirmed that the cleaved LGG-1(G116A), but not the truncated LGG-1(G116AG117\*), is functional for the early step of autophagosome biogenesis. Moreover, RNAi depletion demonstrated that the function of LGG-1(G116A) in aggrephagy pathway was dependent on UNC-51/Ulk1 and the scaffolding protein EPG-2 (Supplementary data and *Figure 7—figure supplement 1*).

Quantification of SEPA-1::GFP in late embryo showed that *lgg-1(G116A)* mutant was able to perform aggrephagy but not *lgg-1(G116AG117\*)* or *lgg-1(G116AG117A)* mutants (*Figure 7F–K*). However, the elimination was decreased compared to wild-type confirming that LGG-1(G116A) was less efficient for selective cargo degradation.

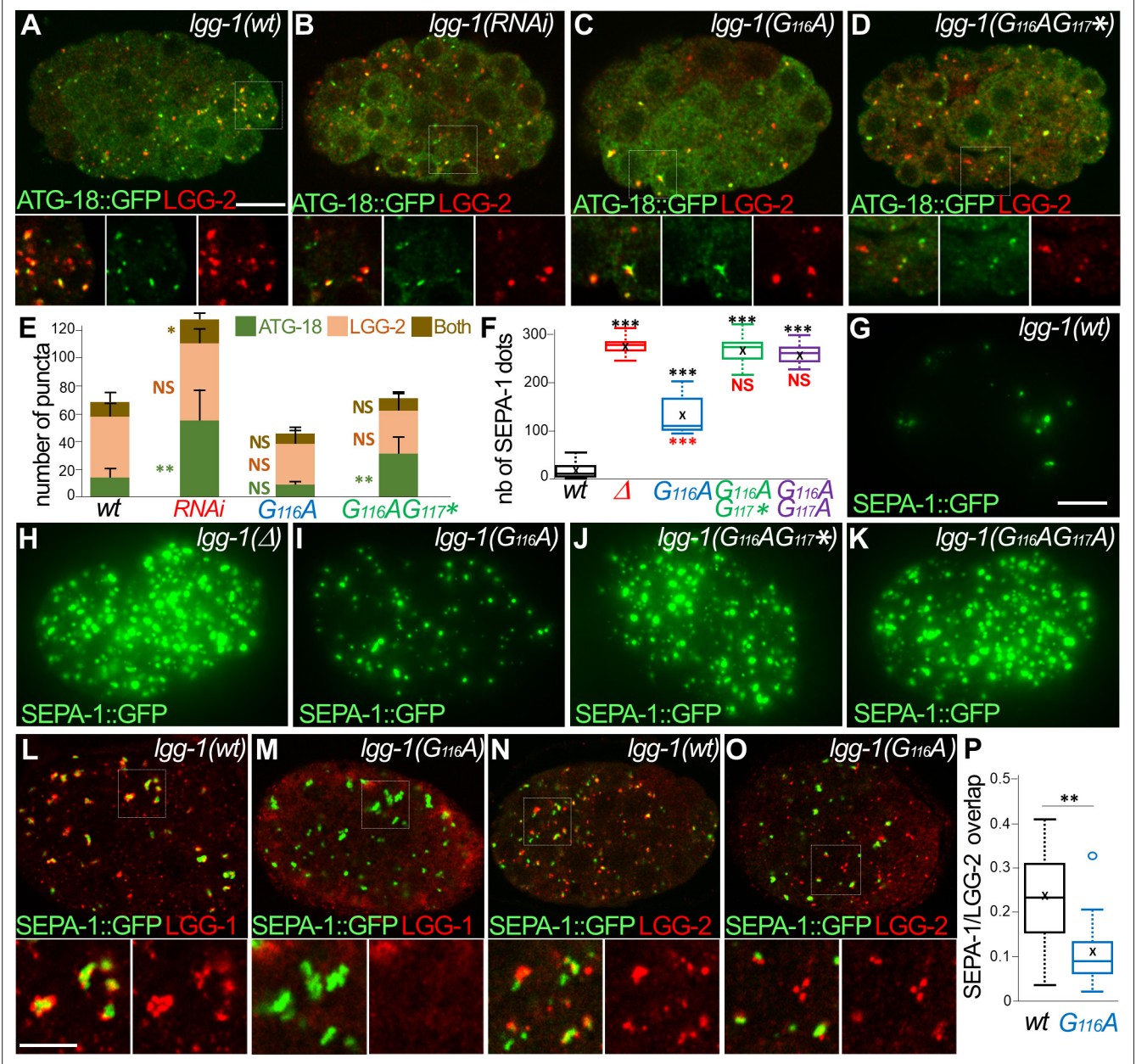

**Figure 7.** The lipidated LGG-1 is involved in the coordination between cargo recognition and autophagosome biogenesis. (**A–E**) Confocal images of ATG-18::GFP (green) and LGG-2 (red) immunofluorescence in *wild-type* (**H**), *lgg-1(RNAi)* (**I**), *lgg-1(G116A)* (**J**), *lgg-1(G116AG117\*)* (**K**) 100 cells embryos. Insets are twofold magnification of the white boxed regions. (**E**) Compared to ATG-18 puncta the number of colocalization is decreased in *lgg-1(RNAi)* (*P*-value <0.05) and *lgg-1(G116AG117\*)* (p-value*<0.001) but not *lgg-1(G116A)* embryos (mean + SD, n=10, 10, 10, 10; Kruskal Wallis p-value*<0.05**<0.01). (**F–K**) Quantification (**F**) and maximum projections of epifluorescence images of the aggrephagy cargo SEPA-1::GFP in 1.5 fold embryos for *wild-type* (**G**), *lgg-1(Δ)* (**H**), *lgg-1(G116A)* (**I**), *lgg-1(G116AG117\*)* (**J**) and *lgg-1(G116AG117A)* (**K**). Boxplots of SEPA-1::GFP dots (n=10) (**F**) indicate that the degradation is stronger in *lgg-1(G116A)* embryos than in *lgg-1(RNAi)*, *lgg-1(G116AG117\*)* and *lgg-1(G116AG117A)* but weaker than in *wt*. (**L–P**) Confocal images of SEPA-1::GFP (green) and LGG-1 (**L, M**) or LGG-2 (**N, O**) (red) immunofluorescence in *wild-type* (**L, N**) and *lgg-1(G116A)* (**M, O**) 100 cells embryos. Insets are 2.5-fold magnification of the white boxed regions. In *lgg-1(G116A)* embryos LGG-2-positive/ LGG-1-negative autophagosomes are detected close to SEPA-1::GFP cargos but with a decreased overlap. (**P**) Box-plots of the overlap between green and red pixels (Manders coefficient) in *wild-type* and *lgg-1(G116A)* (n=11, 13; Mann-Whitney test **<0.01). Scale bar is 10 µm (**A–K**) or 5 µm (**L- O**).

The online version of this article includes the following source data and figure supplement(s) for figure 7:

**Source data 1.** Folder containing original microscopy pictures and quantification data shown in *Figure 7*.

**Figure supplement 1.** LGG-1(G116A) function in aggrephagy is dependent on UNC-51 and EPG-2.

**Figure supplement 1—source data 1.** Folder containing original microscopy pictures and quantification data shown in *Figure 7—figure supplement 1*.

Finally, the interactions between cargoes and autophagosomes were studied in *lgg-1(G116A)* mutant and wild-type embryos by analyzing the colocalization between SEPA-1 and LGG-1 or LGG-2. In wild-type embryos, immunofluorescence analyses showed the presence of LGG-1 and LGG-2 autophagosomes in contact with SEPA-1 aggregates (*Figure 7L and N*). In *lgg-1(G116A)* embryos, LGG-2 positive autophagosomes were observed but no LGG-1 dots, in line with the absence of lipidation (*Figure 7M and O*). A part of LGG-2 puncta was present close to SEPA-1 aggregates, however, they were less numerous and the overlap between LGG-2 and SEPA-1 signals was weaker (*Figure 7P*). These data suggested that LGG-1(G116A) was able to maintain the function of LGG-1 for initiation and extension of autophagosomes but was partially deficient for cargo sequestering.

Altogether, the analyses of LGG-1(G116A) indicate that many of the functions of LGG-1 in autophagy can be achieved by the cleaved, non-lipidated form I. However, the lipidation of LGG-1 appears to be important for the coordination between cargo recognition and autophagosome biogenesis and for the correct degradation of the autophagosome.

## Discussion

The most surprising result of this study is the discovery that LGG-1(G116A) is functional for many autophagy processes, covering physiological or stress conditions and selective or bulk autophagy. To our knowledge, it is the first report demonstrating that different autophagy processes are fully achieved in vivo in a non-lipidated LC3/GABARAP mutant. In cultured cells, an elegant CRISPR strategy allowed to knock out together the six LC3/GABARAP homologs, but point mutations of the conserved glycine have not been reported (*Nguyen et al., 2016*). Most of the studies on the terminal glycine used transgenic overexpression constructs (*Chen et al., 2007*; *Kabeya et al., 2004*). Interestingly, one study reported that part of the autophagy functions of GABARAPL1 is independent of its lipidation (*Poillet-Perez et al., 2017*). Several studies have used mutations in the conjugation machinery (Atg3, Atg5, Atg7) or the Atg4 protease to analyze the role of the form I (*Hill et al., 2019*; *Hirata et al., 2017*; *Nishida et al., 2009*; *Ohnstad et al., 2020*; *Vujić et al., 2021*). A non-canonical autophagy has been reported in *Atg5*, *Atg7* mutants (*Nishida et al., 2009*), but blocking the conjugation system presumably affects all LC3/GABARAP homologs. Moreover, the presence of four homologs of Atg4 in mammals, which specificity versus LC3/GABARAP is unknown, and the dual role in the cleavage of the precursor and the delipidation entangle the analysis of the phenotypes.

Our data show no evidence for an intrinsic function of the LGG-1 precursor but the importance of its active cleavage. This finding is not surprising because in many species the Atg8 precursor is not detected, suggesting that the cleavage occurs very soon after or even during translation. Moreover, phylogenetic analyses of LC3/GABARAP show no conservation in sequence and length of the C-terminus but the presence of at least one residue after the conserved G116. The hypothesis of a selective constraint on the cleavage but not on the C-terminus sequence could explain the persistence of a precursor form. Further studies are necessary to clarify the precise implication of the di-glycine G116G117 in the process.

Albeit a similar sequence, the difference of functionality between the cleaved LGG-1(G116A) and the truncated LGG-1(G116AG117*) suggests that the cleavage allows a first level of specificity for LGG-1 functions. The normal development of *lgg-1(G116AG117*)* animals is the first evidence that LGG-1 function in development relies on the cleavage but is independent of autophagy and conjugation. Our results could explain the embryonic lethality reported upon depletion of the two Atg4 homologs precursors in *C. elegans* (*Wu et al., 2012*). While the cleavage is sufficient for developmental functions, autophagy functions of LGG-1 form I seem to require a further modification to be efficient. Our data suggest that this modification is dependent on and possibly associated to the cleavage. The presence of a new minority band for LGG-1(G116A) could reflect an intermediary transient processing state but should not correspond to a functional form because it was also detected for LGG-1(G116AG117*) and LGG-1(G116AG117A).

Our observations in yeast also support an autophagy independent function of Atg8 form I in vacuolar shaping. Non-autophagic functions for LC3/GABARAP have been identified in yeast and higher eukaryotes (*Ishii et al., 2019*; *Liu et al., 2018*; *Schaaf et al., 2016*; *Wesch et al., 2020*), but the roles of the cytosolic forms are poorly documented especially in the context of the development. The two Atg8 homologs of *Drosophila* are involved in several developmental processes independently of canonical lipidation (*Chang et al., 2013*) or autophagy (*Jipa et al., 2020*). They are highly similar and

both correspond to GABARAP homologs (*Manil-Ségalen et al., 2014*). It is possible that duplication of Atg8 during evolution allowed the acquisition of specific developmental functions by GABARAP proteins but reports in apicomplex parasites (*Lévêque et al., 2015*; *Mizushima and Sahani, 2014*) rather support a non-autophagy ancestral function of Atg8.

The major goal of this study was to bring new insights concerning the implication of LGG-1 form I in various steps of autophagy. Numerous studies identified interacting partners of Atg8/LC3/GABARAP family during autophagy but its mechanistic function for autophagosome biogenesis is still debated. In yeast, the amount of Atg8 regulates the level of autophagy and controls phagophore expansion, but is mainly released from the phagophore assembly site during autophagosome formation (*Xie et al., 2008*). In vitro studies using liposomes or nanodiscs suggested that Atg8 is a membrane-tethering factor and promotes hemifusion (*Nakatogawa et al., 2007*), membrane tubulation (*Knorr et al., 2014*), or membrane-area expansion and fragmentation (*Maruyama et al., 2021*). Another study showed that Atg8–PE assembles with Atg12–Atg5-Atg16 into a membrane scaffold that is recycled by Atg4 (*Kaufmann et al., 2014*). A similar approach with LGG-1 supports a role in tethering and fusion activity (*Wu et al., 2015*). In vivo, the functions of these proteins could depend on their amount, their posttranslational modifications, and the local composition of the membrane. For instance, an excess of lipidation of the overexpressed LGG-1 form I mediates the formation of enlarged protein aggregates and impedes the degradation process (*Wu et al., 2015*). A recent report showed that the phosphorylation of LC3C and GABARAP-L2 impedes their binding to ATG4 and influences their conjugation and de-conjugation (*Herhaus et al., 2020*).

Our genetic data suggest that form I of LGG-1 is sufficient for initiation, elongation, and closure of autophagosomes but that lipidated LGG-1 is important for the cargo sequestering and the dynamics of degradation. However, the partial redundancy with LGG-2 is presumably an important factor during these processes. If the main functions of LGG-1 reside in its capacity to bind multiple proteins, the localization to autophagosome membrane through lipidation is an efficient but not unique way to gather cargoes and autophagy complexes. Furthermore, the possibility that non-positive LGG-1/LGG-2 autophagosomes could mediate cargo degradation questions the use of Atg8/GABARAP/LC3 family as a universal marker for autophagosomes. Overall, our results confirm the high level of plasticity and robustness of autophagosome biogenesis.

# Materials and methods

**Key resources table**

| Reagent type (species) or resource | Designation | Source or reference | Identifiers | Additional information |
|---|---|---|---|---|
| Gene (*C. elegans*) | *lgg-1* | Wormbase | WBGene00002980 | |
| Strain, strain background (*C. elegans*) | N2 | CGC | | *Wild-type strain* |
| Genetic reagent (*C. elegans*) | DA2123 | CGC | | *adIs2122[gfp::lgg-1;rol-6(su1006)]* |
| Genetic reagent (*C. elegans*) | GK1057 | *Sato and Sato, 2011* | | *Pspe-11-hsp-6::GFP* |
| Genetic reagent (*C. elegans*) | HZ455 | CGC | | *him-5(e1490) V; bpIs131[sepa-1::gfp]* |
| Genetic reagent (*C. elegans*) | HZ1685 | CGC | | *atg-4.1(bp501)* |
| Genetic reagent (*C. elegans*) | MAH247 | CGC | | *sqIs25[atg-18 p::atg-18::gfp +rol-6(su1006) ]* |
| Genetic reagent (*C. elegans*) | RD202 | Legouis lab | | *Is202[unc-119(ed3)III;plgg-1::GFP::LGG-1 G->A]* |
| Genetic reagent (*C. elegans*) | lgg-1(Δ) | Mitani lab | NBRP: tm3489 | *lgg-1(tm3489)* |
| Genetic reagent (*C. elegans*) | lgg-2(tm5755) | Mitani lab | NBRP: tm5755 | *lgg-2(tm5755)* |

*Continued on next page*

*Continued*

| Reagent type (species) or resource | Designation | Source or reference | Identifiers | Additional information |
|---|---|---|---|---|
| Genetic reagent (*C. elegans*) | RD363; lgg-1(Δ112–123) | This paper | | *lgg-1(pp22)dpy-10(pp157)* Legouis lab |
| Genetic reagent (*C. elegans*) | RD367; lgg-1(G116A) | This paper | | *lgg-1(pp65[G116A])* Legouis lab |
| Genetic reagent (*C. elegans*) | RD368; lgg-1(Δ100–123) | This paper | | *lgg-1(pp66)* Legouis lab |
| Genetic reagent (*C. elegans*) | RD420; lgg-1(G116AG117*) | This paper | | *lgg-1(pp141[G116AG117stop])* Legouis lab |
| Genetic reagent (*C. elegans*) | RD421; lgg-1(G116AG117A) | This paper | | *dpy-10(pp163)lgg-1(pp142[G116AG117A])* Legouis lab |
| Genetic reagent (*C. elegans*) | RD425 | This paper | | *dpy-10(pp163)lgg1(pp142)/+; SEPA-1::gfp* Legouis lab |
| Genetic reagent (*C. elegans*) | RD435 | This paper | | *lgg-1(pp141[G116AG117stop]); atg-18 p::atg-18::gfp +rol-6(su1006)* Legouis lab |
| Genetic reagent (*C. elegans*) | RD436 | This paper | | *lgg-1(pp65[G116A]); atg-18 p:: atg-18::gfp +rol-6(su1006)* Legouis lab |
| Genetic reagent (*C. elegans*) | RD440 | This paper | | *lgg-1(pp141[G116AG117stop]); lgg-2(tm5755)* Legouis lab |
| Genetic reagent (*C. elegans*) | RD446 | This paper | | *lgg-1(pp65[G116A]); lgg-2(tm5755)* Legouis lab |
| Genetic reagent (*C. elegans*) | RD447 | This paper | | *lgg-1(tm3489); atg-18 p::atg-18::gfp +rol-6(su1006)* Legouis lab |
| Genetic reagent (*C. elegans*) | RD448 | This paper | | *lgg-1(pp65[G116A]); SEPA-1::gfp* Legouis lab |
| Genetic reagent (*C. elegans*) | RD449 | This paper | | *lgg-1(pp141[G116AG117stop]); SEPA-1::gfp* Legouis lab |
| Genetic reagent (*C. elegans*) | RD450 | This paper | | *lgg-1(tm3489)II; SEPA-1::gfp* Legouis lab |
| Strain, strain background (*S. cerevisiae*) | BY4742 | Euroscarf | | *Mat alpha ura3Δ0, his3Δ1, leu2Δ0, lys2Δ0* |
| Genetic reagent (*S. cerevisiae*) | OC513 | YKO collection | | BY4742, *atg1::KanMX4* |
| Genetic reagent (*S. cerevisiae*) | OC612 | YKO collection | | BY4742, *atg8::KanMX4* |
| Genetic reagent (*S. cerevisiae*) | OC608-OC609 | This paper | | BY4742, *atg8G116A* Legouis lab |
| Genetic reagent (*S. cerevisiae*) | OC610-OC611 | This paper | | BY4742, *atg8G116A-R117** Legouis lab |
| Genetic reagent (*S. cerevisiae*) | OC613 | This paper | | BY4742, *pho8::pho8Δ60-URA3KL* Legouis lab |
| Genetic reagent (*S. cerevisiae*) | OC614 | This paper | | BY4742, *atg1::KanMX4, pho8:: pho8Δ60-URA3KL* Legouis lab |
| Genetic reagent (*S. cerevisiae*) | OC615 | This paper | | BY4742, *atg8::KanMX4, pho8:: pho8Δ60-URA3KL* Legouis lab |
| Genetic reagent (*S. cerevisiae*) | OC616-OC617 | This paper | | BY4742, *atg8G116A, pho8:: pho8Δ60-URA3KL* Legouis lab |
| Genetic reagent (*S. cerevisiae*) | OC618-OC619 | This paper | | BY4742, *atg8G116A-R117*, pho8:: pho8Δ60-URA3KL* Legouis lab |
| Strain strain background (*E. coli*) | OP50 | CGC | | see Material and Methods |
| Genetic reagent (*E. coli*) | JA-C32D5.9 | Open Biosystem | | *lgg-1* RNAi feeding bacterial clone |

*Continued on next page*

*Continued*

| Reagent type (species) or resource | Designation | Source or reference | Identifiers | Additional information |
|---|---|---|---|---|
| Genetic reagent (*E. coli*) | JA-C56C10.12 | Open Biosystem | | *epg-5* RNAi feeding bacterial clone |
| Genetic reagent (*E. coli*) | JA-Y55F3AM.4 | Open Biosystem | | *atg-3* RNAi feeding bacterial clone; |
| Genetic reagent (*E. coli*) | JA-M7.5 | Open Biosystem | | *atg-7* RNAi feeding bacterial clone |
| Genetic reagent (*E. coli*) | JA-W03C9.3 | Open Biosystem | | *rab-7* RNAi feeding bacterial clone |
| Genetic reagent (*E. coli*) | JA- Y39G10AR.10 | Open Biosystem | | *epg-2* RNAi feeding bacterial clone |
| Sequence-based reagent | CrRNA(s) | *Paix et al., 2015* | | *dpy-10* : 5'GCUACCAUAGGCACCACGAGGU UUUAGAGCUAUGCUGUUUUG3' |
| Sequence-based reagent | CrRNA(s) | This paper | | *lgg-1* Legouis lab 5'UACAGUGACGAAAGUGUG UAGUUUUAGAGCUAUGCUGUUUUG3' |
| Sequence-based reagent | Repair template | *Paix et al., 2015*; | | *dpy-10* : 5'CACTTGAACTTCAATACGGCAAGATG AGAATGACTGGAAACCGTACCGCATGCGG TGCCTATGGTAGCGGAGCTTCACATGGC TTCAGACCAACAGCCTAT3' |
| Sequence-based reagent | Repair template | This paper | | *lgg-1* (G116A): Legouis lab 5'CTTTACATCGCGTACAGTGACGAAAGT GTCTACGCCGGAGAGGTCGAAAAGAAG GAATAAAGTGTCATGTAT3' |
| Sequence-based reagent | Repair template | This paper | | *lgg-1* (G116AG117 *): Legouis lab 5'TTCCTTTACATCGCCTACAGTGACGAAAGT GTGTACGCCTAAGAATTCGAAAAGAAGGAAT AAAGTGTCATGTATTATCCG3' |
| Sequence-based reagent | Repair template | This paper | | *lgg-1* (G116AG117A): Legouis lab 5'TTCCTTTACATCGCCTACAGTGACGAAAGT GTGTACGCCGCAGAGGTCGAAAAGAAGGA ATAAGAATTCAGTGTCATGTATTAT CCGCCGACGAATGTGTATAC3' |
| Sequence-based reagent | Universal tracrRNA | Dharmacon GE | U-002000–05 | 5'AACAGCAUAGCAAGUUAAAAUAAGGCU AGUCCGUUAUCAACUUGAAAAAGUGGC ACCGAGUCGGUGCUUUUUUU3' |
| Peptide, recombinant protein | *S. pyogenes* Cas9 | Dharmacon | CAS11201 | Edit-R Cas9 Nuclease Protein, 1000 pmol |
| Antibody | anti-LGG-1 (rabbit polyclonal) | *Springhorn and Hoppe, 2019* | | Ab#3 WB (1:3000) |
| Antibody | anti-LGG-1 (rabbit polyclonal) | *Al Rawi et al., 2011* | | Ab#1 WB (1:200) IF(1:100) |
| Antibody | anti-LGG-2 (rabbit polyclonal) | *Manil-Ségalen et al., 2014* | | WB (1:200) IF (1:200) |
| Antibody | anti-Tubulin (mouse monoclonal) | Sigma | 078K4763 | WB (1:1000) |
| Antibody | anti-SEL-1 (rabbit polyclonal) | Hoppe's lab | | WB (1:8000) |
| Antibody | anti-CDC-48.1 (rabbit polyclonal) | Hoppe's lab | | WB (1:5000) |
| Antibody | Anti-Rabbit HRP (goat polyclonal) | Promega | W401B | WB (1:5000) |
| Antibody | Anti-mouse HRP (goat polyclonal) | Promega | W4021 | WB (1:10,000) |
| Antibody | anti-GABARAP (rabbit polyclonal) | Chemicon | AB15278 | IF (1:200) |
| Antibody | anti-GFP (mouse monoclonal) | Roche | 1814460 | IF (1:250) |

*Continued on next page*

*Continued*

| Reagent type (species) or resource | Designation | Source or reference | Identifiers | Additional information |
|---|---|---|---|---|
| Antibody | anti-mouse IgG Alexa Fluor488 (goat polyclonal) | Molecular Probes | A11029 | IF (1:500 to 1:1000) |
| Antibody | anti-rabbit IgG Alexa Fluor488 (goat polyclonal) | Molecular Probes | A110034 | IF (1:500 to 1:1000) |
| Antibody | anti-rabbit IgG Alexa Fluor568 (goat polyclonal) | Sigma-Aldrich | A11036 | IF (1:500 to 1:1000) |
| Antibody | anti-GFP (rabbit polyclonal) | Abcam | ab6556 | (Immunogold 1:10) |
| Antibody | anti-rabbit IgG (goat polyclonal) | Biovalley | 810.011 | Coupled to 10 nm colloidal gold particles (Immunogold 1:20) |
| Chemical compound, drug | EPON | Agar Scientific | R1165 | see Materials and methods |
| Chemical compound, drug | lead citrate | Sigma-Aldrich | 15326 | see Materials and methods |
| Chemical compound, drug | LRWHITE | Electron Microscopy Sciences | 14381 | see Materials and methods |
| Peptide, recombinant protein | LC3 traps | *Quinet et al., 2022* | | Molecular traps for LGG-1 |
| Commercial assay or kit | Super Signal Pico Chemiluminescent Substrate | Thermo Scientific | 34579 | see Materials and methods |
| Commercial assay or kit | NuPAGE 4%-12% Bis- Tris gel | Life Technologies | NP0321BOX | see Materials and methods |
| Software, algorithm | ImageJ | http://imagej.nih.gov/ij | | see Materials and methods |
| Software, algorithm | Fidji | https://fiji.sc/ | | see Materials and methods |
| Software, algorithm | Prism | GraphPad | | see Materials and methods |
| Software, algorithm | R software | https://www.r-project.org/ | | see Materials and methods |
| Software, algorithm | Crispr | http://Crispr.mit.edu | | see Materials and methods |
| Software, algorithm | Crispor | http://crispor.org | | see Materials and methods |
| Other | MitoTracker Red CMXRos | Molecular Probes | M7512 | see Materials and methods |

Further information and requests for resources and reagents should be directed to the corresponding author, Renaud Legouis (renaud.legouis@i2bc.paris-saclay.fr).

## *C. elegans* culture and strains

Nematode strains were grown on nematode growth media [for 500 ml H2O: 1.5 g NaCl (Sigma-Aldrich, 60142), 1.5 g bactopeptone (Becton-Dickinson, 211677)**,** 0.5 ml cholesterol (Sigma-Aldrich, C8667) 5 mg/ml, 10 g bacto agar (Becton-Dickinson, 214010) supplemented with 500 µl CaCl2 (Sigma-Aldrich, C3306) 1 M, 500 µl MgSO4 (Sigma-Aldrich, M5921) 1 M, 10 ml KH2PO4 (Sigma-Aldrich, P5655) 1 M, 1650 µl K2HPO4 (Sigma-Aldrich, 60356) 1 M] and fed with *Escherichia coli* strain OP50.

## CRISPR-Cas9

A CRISPR-Cas9 approach optimized for *C. elegans* was used, based on a *dpy-10* co-CRISPR protocol (*Paix et al., 2015*). All reagents are in 5 mM Tris-HCl pH 7.5. Crispr.mit.edu and CRISPOR (http://crispor.org) web tools were used to choose a Cas9 cleavage site (NGG) close to the edit site, the best sequence of the crRNAs (50 to 75% of GC content), and for off-target prediction. 1 µL of CrRNA(s) (8 µg/µL or 0.6 nmole/µL) and repair template(s) (1 µg/µL) designed for *lgg-1* and *dpy-10* genes were mixed with 4.1 µL of *S. pyrogenes* Cas9 (20 pmole/µL) and 5 µL of universal tracrRNA (4 µg/µL 4 µg/µL or 0.17 nmol/µL molarity) in 0.75 µL Hepes (200 mM) 0.5 µL KCl (1 M) and water up to 20 µL. The mix was heated for 10 min at 37 °C and injected in the gonad of young adult hermaphrodites. Progenies of injected animals were cloned and genotyped by PCR. Mutants were outcrossed three times and *lgg-1* gene was sequenced to check for the specific mutations.

## Nematode starvation and lifespan

For starvation experiments, adult hermaphrodites were bleached to obtain synchronized L1 larvae. L1 were incubated in 0.5 mL sterilized M9 at 20 °C on spinning wheel. At each time point, an aliquot from each sample tube was placed on a plate seeded with *E. coli* OP50. The number of worms surviving to adulthood was counted 2 or 4 days after. Life span was performed on more than 100 animals for each genotype with independent duplicates and analyzes using Kaplan-Meier method and Log-Rank (Mantel-Cox) test.

## RNA mediated interference

RNAi by feeding was performed as described (*Kamath et al., 2003*). Fourth-larval stage (L4) animals or embryos were raised onto 1 mM isopropyl-D-β-thiogalactopyranoside (IPTG)-containing nematode growth media (NGM) plates seeded with bacteria (*E. coli* HT115[DE3]) carrying the empty vector L4440 (pPD129.36) as a control or the bacterial clones from the J. Ahringer library, Open Biosystem.

## Western blot and cellular fractionation

The worms were collected after centrifugation in M9 and then mixed with the lysis buffer described previously (*Springhorn and Hoppe, 2019*) (25 mM tris-HCl, pH7.6; 150 mM NaCl; 1 mM ethylenediaminetetraacetic acid (EDTA) 1% Triton X-100; 1% sodium deoxycholate (w/v); 1% SDS (w/v)) containing glass beads (Sigma-Aldrich 425–600 µm G8772100G). They were then denatured using Precellys 24 machine at 6000 rpm with incubation for about 5 min twice to cool down the sample. The protein extracts are then centrifuged at 15,000 rpm and separated on a NuPAGE 4%-12% Bis-Tris gel (Life Technologies, NP0321BOX). The non-specific sites are then blocked after the incubation for one hour with PBS Tween 0.1% (Tris Base NaCl, Tween20) BSA 2%. Blots were probed with anti-LGG-1 (1:3000 rabbit Ab#3 *Springhorn and Hoppe, 2019* or 1:200 Ab#1 *Al Rawi et al., 2011*), anti-LGG-2 (1:200 rabbit), anti-Tubulin (1:1000 mouse; Sigma, 078K4763), anti-SEL-1 (1:8000, rabbit), anti-CDC-48.1 (1:5000, rabbit) and revealed using HRP-conjugated antibodies (1: 5000 promega W401B and 1:10,000 promega W4021) and the Super Signal Pico Chemiluminescent Substrate (Thermo Fisher Scientific, 34579). Signals were revealed on a Las3000 photoimager (Fuji) and quantified with Image Lab software. For cellular fractionation, 4000 age-synchronized worms (day 1 of adulthood) were collected from NGM/OP-50 plates, washed three times with M9 buffer and transferred to NGM plates without OP-50 to induce starvation. Worms were starved at 20 °C for 7 hr, and then transferred to fractionation buffer (50 mM Tris-HCl pH 7.4, 150 mM NaCl, 1 mM DTT, 1 mM PMSF, and protease inhibitor cocktail). For cell lysis, worms were homogenized 50 times using a Dounce homogenizer and sonicated for 20 s at 60% amplitude. Cell lysates were centrifuged at 500 RCF and 4 °C for 5 min to remove cell debris and the nuclear fraction. The supernatant was centrifuged again at 20,000 RCF and 4 °C for 90 min to separate soluble (cytosolic) and insoluble (membrane) fractions. Supernatant and pellet were separated and the pellet was resuspended in 150 µL of fractionation buffer. From this step, input samples were prepared for Western blot analysis. Subsequently, 30 µL of the pellet sample was mixed with 3 µL each of fractionation buffer, 3 µL 5 M NaCl, and 3 µL Triton X-100. Treated pellet samples were incubated on ice for 1 hour and then centrifuged at 20,000 RCF and 4 °C for 60 min. The resulting supernatants and pellets were again separated and analyzed by Western blotting.

## Immunofluorescence and light microscopy

Fifty adult hermaphrodites were cut to release the early embryos on a previously poly-L-lysinated slide (0.1%). Late embryos were deposited using a flattened platinum wire and bacteria as glue. Embryos were prepared for immunofluorescence staining by freeze-fracture and methanol fixation 30 min at –20 °C, incubated 40 min in 0.5% Tween, 3% BSA, PBS solution, and washed twice 30 min in 0.5% Tween PBS solution. Incubation overnight at 4 °C overnight with the primary antibodies anti-LGG-1(rabbit 1:100) anti-GABARAP (rabbit 1:200) (1: 200), anti-LGG-2 (rabbit 1:100) was followed by two washes, 2 hr incubation at room temperature with the secondary antibodies, Alexa488 and Alexa 568 (1: 1000), and two washes. Embryos were mounted in DABCO and imaged on an AxioImagerM2 microscope (Zeiss) equipped with Nomarski optics, coupled to a camera (AxioCam506mono) or a confocal Leica TCS SP8 microscope with serial z sections of 0.5–1 µm. Images were analyzed, quantified and processed using ImageJ or Fiji softwares.

For live imaging samples were mounted on a 2% agarose pad and larvae were immobilized by 40 mM sodium azide. For MitoTracker staining, adult worms were transferred to NGM agar plates containing 3.7 μM of Red CMXRos (Molecular Probes, Invitrogen) and incubated for overnight in the dark.

## Electronic microscopy

One-day adults were transferred to M9 20% BSA (Sigma-Aldrich, A7030) on 1% phosphatidylcholine (Sigma-Aldrich) pre-coated 200 μm deep flat carriers (Leica Biosystems), followed by cryo-immobilization in the EMPACT-2 HPF apparatus (Leica Microsystems; Vienna Austria) as described (*Jenzer et al., 2019*). Cryo-substitution was performed using an Automated Freeze-substitution System (AFS2) with integrated binocular lens, and incubating chamber (Leica Microsystems; Wetzlar, Germany) with acetone. Blocks were infiltrated with 100% EPON, and embedded in fresh EPON (Agar Scientific, R1165). Ultrathin sections of 80 nm were cut on an ultramicrotome (Leica Microsystems, EM UC7) and collected on a formvar and carbon-coated copper slot grid (LFG, FCF-2010-CU-50). Sections were contrasted with 0,05% Oolong tea extract (OTE) for 30 min and 0.08 M lead citrate (Sigma-Aldrich, 15326) for 8 min. Sections were observed with a Jeol 1400 TEM at 120 kV and images acquired with a Gatan 11 Mpixels SC1000 Orius CCD camera.

## Affinity purification of LGG-1

One mg of total proteins from *C. elegans* lysate were incubated on ice 10 min in 800 μL of TUBE lysis buffer [50 mM sodium fluoride, 5 mM tetra-sodium pyrophosphate, 10 mM β-glyceropyrophosphate, 1% Igepal CA-630, 2 mM EDTA, 20 mM $Na_2HPO_4$, 20 mM $NaH_2PO_4$, and 1.2 mg/ml complete protease inhibitor cocktail (Roche, Basel, Switzerland)] supplemented with 200 μg of purified LC3 traps or GST control (*Quinet et al., 2022*). After cold centrifugation at 16,200 *g* for 30 min, supernatant was harvested and added to 400 μl of prewashed glutathione-agarose beads (Sigma), and incubated for 6 hr rotating at 4 °C. Beads were centrifugated at 1000 *g* for 5 min at 4 °C (Beckman Coulter Microfuge 22 R, Fullerton, CA, USA), washed five times using 10 column volumes of PBS-tween 0.05%. Elution was done in 100 μL of (Tris pH7.5, 150 mM NaCl, 1% Triton, 1% SDS) at 95 °C during 10 min, and supernatant was harvested.

## Mass spectrometry

Protein samples affinity purification were prepared using the single-pot, solid-phase-enhanced sample-preparation (SP3) approach as described (*Hughes et al., 2019*). Samples were mixed with 10 μl of 10 μg/μl solution of Sera-Mag SpeedBeadsTM hydrophilic and hydrophobic magnetics beads (GE healthcare, ref 45152105050250 and 65152105050250) with a bead to sample ratio of 10:1. After a binding step in 50% ethanol in water, and three successive washes with 80% ethanol in water, sample were digested with 100 μl of a 5 ng/μl sequencing grade modified trypsin solution (PROMEGA). Fifty μl of Trypsin-generated peptides were vacuum dried, resuspended in 10 μl of loading buffer (2% acetonitrile and 0.05% Trifluoroacetic acid in water) and analyzed by nanoLC-MSMS using a nanoElute liquid chromatography system (Bruker) coupled to a timsTOF Pro mass spectrometer (Bruker). Briefly, peptides were loaded on an Aurora analytical column (ION OPTIK, 25cm x75μm, C18, 1.6 μm) and eluted with a gradient of 0–35% of solvent B for 100 min. Solvent A was 0.1% formic acid and 2% acetonitrile in water, and solvent B was 99.9% acetonitrile with 0.1% formic acid. MS and MS/MS spectra were recorded and converted into mgf files. Proteins identification were performed with Mascot search engine (Matrix science, London, UK) against a database composed of all LGG-1 sequences including the wild-type and mutant sequences. Database searches were performed using semi-trypsin cleavage specificity with five possible miscleavages. Methionine oxidation was set as variable modification. Peptide and fragment tolerances were set at 15 ppm and 0.05 Da, respectively. A peptide mascot score threshold of 13 was set for peptide identification. C-terminal peptides were further validated manually.

## Quantification and statistical analysis

All experiments were done at least three times. All data summarization and statistical analyses were performed by using either the GraphPad-Prism or the R software (https://www.r-project.org/). The Shapiro-Wilk's test was used to evaluate the normal distribution of the values and the Hartley Fmax

test for similar variance analysis. Data derived from different genetic backgrounds were compared by Student t test, Anova, Kruskal-Wallis or Wilcoxon-Mann-Whitney tests. The Fisher's exact test was used for nominal variables. Longevity was assessed using Log-Rank (Mantel-Cox) test. Error bars are standard deviations and boxplot representations indicate the minimum and maximum, the first (Q1/25th percentile), median (Q2/50th percentile) and the third (Q3/75th percentile) quartiles. NS (Not Significant) $p>0.05$; * $0.05>p > 0.01$, **$0.01>p > 0.001$, *** $0.001>p > 0.0001$ and **** $p<0.0001$. Exact values of n and statistical tests used can be found in the figure legends.

## Acknowledgements

The authors thank Fulvio Reggiori for yeast plasmids, and the Caenorhabditis Genetic Center, which is funded by the NIH National Center for Research Resources (NCRR), for strains. We are grateful to Laïla Sago and Virginie Redeker for help with mass spectrometry. The present work has benefited from the facilities and expertise of the I2BC proteomic platform (Proteomic-Gif, SICaPS) supported by IBiSA, Ile de France Region, Plan Cancer, CNRS and Paris-Sud University as well as the core facilities of Imagerie-Gif, member of IBiSA, supported by "France-BioImaging" and the Labex "Saclay Plant Science".

This work was funded by the Deutsche Forschungsgemeinschaft (DFG, German Research Foundation) under Germany´s Excellence Strategy (EXC 2030) 390661388 and by the European Research Council (ERC-CoG-616499) to TH.This work was supported by the Agence Nationale de la Recherche (project EAT, ANR-12-BSV2-018), the Association pour la Recherche contre le Cancer (SFI20111203826) and the Ligue contre le Cancer (MM). RoL received a fellowship from Fondation pour la Recherche Médicale.

## Additional information

### Funding

| Funder | Grant reference number | Author |
|---|---|---|
| Fondation pour la Recherche Médicale | ECO20170637554 | Romane Leboutet |
| Agence Nationale de la Recherche | project EAT | Renaud Legouis |
| Fondation ARC pour la Recherche sur le Cancer | SFI20111203826 | Renaud Legouis |
| Ligue Contre le Cancer | M29506 | Renaud Legouis |
| Agence Nationale de la Recherche | ANR-12-BSV2-018 | Renaud Legouis |
| Deutsche Forschungsgemeinschaft | (EXC 2030) 390661388 | Thorsten Hoppe |
| European Research Council | ERC-CoG-616499 | Thorsten Hoppe |

The funders had no role in study design, data collection and interpretation, or the decision to submit the work for publication.

### Author contributions

Romane Leboutet, Formal analysis, Validation, Investigation, Visualization, Methodology, Writing – review and editing; Céline Largeau, Magali Prigent, Formal analysis, Validation, Investigation, Visualization; Leonie Müller, Formal analysis, Investigation, Writing – review and editing; Grégoire Quinet, Resources, Investigation; Manuel S Rodriguez, Resources; Marie-Hélène Cuif, Formal analysis, Validation, Investigation, Visualization, Writing – review and editing; Thorsten Hoppe, Resources, Supervision, Writing – review and editing; Emmanuel Culetto, Conceptualization, Supervision, Investigation, Methodology, Writing – review and editing; Christophe Lefebvre, Supervision, Investigation,

Methodology; Renaud Legouis, Conceptualization, Supervision, Funding acquisition, Validation, Writing - original draft, Project administration, Writing – review and editing

**Author ORCIDs**
Thorsten Hoppe http://orcid.org/0000-0002-4734-9352
Emmanuel Culetto http://orcid.org/0000-0003-4725-2654
Renaud Legouis http://orcid.org/0000-0002-2699-2584

**Decision letter and Author response**
Decision letter https://doi.org/10.7554/eLife.85748.sa1
Author response https://doi.org/10.7554/eLife.85748.sa2

## Additional files

**Supplementary files**
• MDAR checklist

**Data availability**
All data generated or analysed during this study are included in the manuscript and supporting file. Further information and requests for resources and reagents should be directed to the corresponding author, Renaud Legouis (renaud.legouis@i2bc.paris-saclay.fr).

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

# Appendix 1

## LGG-1(wt) and LGG-1(G116A) partially restore the survival to nitrogen starvation of yeast *atg8(Δ)*

Our results demonstrate that the localization of LGG-1 to the membrane is dispensable for autophagy and other functions. To address whether it is a particularity of *C. elegans*, a similar strategy was performed for Atg8 in the yeast *S. cerevisiae*. Atg8 precursor ends with an arginine at position 117 (*Figure 1—figure supplement 3A*) and using mutant proteins expressed from centromeric plasmids the Oshumi lab (*Kirisako et al., 2000*; *Nakatogawa et al., 2012*) has shown that G116 is essential for autophagy. The endogenous *ATG8* was modified by an homologous recombination strategy to generate *atg8(G116A)* and *atg8(G116AR117*)* alleles, and the autophagy flux was assessed using the Pho8Δ60 reporter (*Noda and Klionsky, 2008*). Both *atg8(G116A)* and *atg8(G116AR117*)* mutants were unable to achieve a functional autophagy and behave similarly to *atg8(Δ)* or *atg1(Δ)* null mutants (*Figure 1—figure supplement 3B*). The analysis of nitrogen starvation survival showed that both *atg8(G116A)* and *atg8(G116AR117*)* strains were unable to recover after a 4 days starvation, similarly to *atg1Δ* and *atg8Δ* (*Figure 1—figure supplement 3C*).

The non-autophagy function of *atg8(G116A)* and *atg8(G116AR117*)* mutants was assessed by analyzing the shape of the vacuole (*Banta et al., 1988*). During exponential growth, *atg8(Δ)* cells frequently presented multiple small vacuoles compared to wild-type cells which harbor usually less than 4 vacuoles (*Figure 1—figure supplement 3D, E*). The incidence of defective vacuolar shape decreased in *atg8(G116A)* and *atg8(G116AR117*)* cells indicating that the non autophagy functions were partially maintained. These data suggest that in *S. cerevisiae* the non-autophagy functions of Atg8 are partially independent of its cleavage and conjugation.

*Atg8* mutant was then used to investigate whether the functionality of LGG-1(G116A) in autophagy was restricted to *C. elegans*. The capacity of LGG-1(G116A) to restore nitrogen starvation survival to *atg8(Δ)* mutant cells was compared with LGG-1(wt) and LGG-1(G116AG117*). The corresponding cDNAs were cloned in a centromeric vector and the proteins were expressed in *atg8(Δ)* mutants. The expression of LGG-1(wt) and LGG-1(G116A), but not LGG-1(G116AG117*), improved weakly the nitrogen starvation survival indicating a partial complementation (*Figure 1—figure supplement 3F*).This suggests that the functionality of LGG-1(G116A) is not restricted to *C. elegans,* supporting an intrinsic property of LGG-1 form I.

## LGG-1(G116A) function in aggrephagy is dependent on UNC-51 and EPG-2

Data in yeast and mammals have revealed that Atg8/LC3/GABARAP can interact with Atg1/ULK1 and modify the kinase activity of the ULK1 complex (*Alemu et al., 2012*; *Grunwald et al., 2020*; *Kraft et al., 2012*; *Nakatogawa et al., 2012*). In *C. elegans*, LGG-1 can directly binds UNC-51/ULK1 (*Wu et al., 2015*) and the cargo SEPA-1 (*Zhang et al., 2009*) and could have an early function for initiating aggrephagy (*Lu et al., 2011*). To decipher the function of LGG-1 form I in the induction of autophagy we performed a genetic approach. Using RNAi we depleted UNC-51 and quantified the initiation events in vivo in wild-type, *lgg-1(Δ)*, *lgg-1(G116A)*, and *lgg-1(G116AG117*)* embryos (*Figure 7—figure supplement 1A–J*). As expected, depleting UNC-51 resulted in the decrease of ATG-18 puncta while depleting LGG-1 lead to the increase of ATG-18 intensity and number of puncta. The decrease of ATG-18::GFP puncta after co-depletion of UNC-51 and LGG-1 indicated that LGG-1 functions depends on UNC-51. For *lgg-1(G116A)* animals, a small decrease was observed in the number of puncta compared to the wild-type animals but no change in the total signal of ATG-18::GFP. Moreover, the depletion of UNC-51 further decreased ATG-18::GFP puncta and intensity confirming that LGG-1(G116A) almost behaves like the wild-type LGG-1 for the initiation (*Figure 7—figure supplement 1C, G, I, J*). In *lgg-1(G116AG117*)* embryos, a marked increase of ATG-18 puncta was observed, but contrarily to *lgg-1(Δ)*, the number did not decrease when UNC-51 was depleted (*Figure 7—figure supplement 1D, H, I, J*). These data confirm electron microscopy observations and support a neomorphic function for the truncated LGG-1(G116AG117*), independent of ULK1 complex.

Finally, we analyzed whether the cargoes degradation by LGG-1(G116A) was dependent of the scaffolding protein EPG-2 (*Figure 7—figure supplement 1K–O*). Aggregate-prone proteins are degraded through autophagy in *C. elegans* embryo through liquid-liquid phase separation promoted by the receptor SEPA-1 and regulated by the scaffolding protein EPG-2 (*Zhang et al.,*

*2018*). The depletion of EPG-2 induced the persistence of SEPA-1::GFP aggregates in wild-type and in *lgg-1(G116A)* embryos. This data indicates that the function of LGG-1(G116A) for degrading SEPA-1 is dependent of EPG-2.

## Supplementary Material and Methods

### Immunostaining and electron microscopy

200 µm-deep flat carriers (Leica Biosystems) were incubated few minutes in 1% phosphatidylcholine (Sigma-Aldrich,61755) in chloroform. Young adults were transferred to the carriers containing 20% BSA (Sigma-Aldrich, A7030) in M9 buffer, followed by cryo-immobilization in the EMPACT-2 HPF apparatus (Leica Microsystems) and cryo-substitution with Automated Freeze-substitution System (AFS2, Leica Microsystems). Cryosubtitution medium, composed by 0.1% acetate uranyl in acetone, for 3 days with a slow increase of the temperature from –90°C to –15°C. After several washes of acetone and ethanol at –15 °C, samples were incubated successively in 25% to 100% LRWHITE resin (Electron Microscopy Sciences, 14381) in ethanol, then UV-polymerized 24 hours at –15 °C. 80 nm thin sections were collected on a Nickel 100-mesh grids (Electron Microscopy Sciences, FCF-100-Ni) and immunostained with the immunogold labelling system (IGL, Leica microsystem). Samples were labelled during 1 h with the primary rabbit anti-GFP antibody (Abcam, ab6556; 1:10 dilution in 0.1% BSA in PBS), washed 4 times for 2 min with PBS, and twice for 5 min with 0.1% BSA in PBS. Samples where then labelled during 30 min with the secondary goat anti-rabbit antibody coupled to 10 nm colloidal gold particles (Biovalley, 810.011) at 1:20 dilution in 0.1% BSA in PBS. Sections were contrasted with 2% uranyl acetate (Merck, 8473) for 8 min and 0.08 M lead citrate (Sigma-Aldrich, 15326) for 2 min, and were observed with a Jeol 1400 TEM at 120 kV equipped with a Gatan SC1000 Orius CCD camera (Roper Industries).

### *S. cerevisiae* culture and strains

Yeast cells were grown to log phase in YPD (1% yeast extract, 2% bactopeptone and 2% glucose) or complete synthetic medium (CSM) without uracil or leucine. The reference strain is BY4742. Other strain and genotypes are listed in the Key Resources Table.

### *S. cerevisiae* culture and autophagy assays

The quantitative Pho8Δ60 assay for bulk autophagy, was performed as described (***Noda and Klionsky, 2008***). Cells were grown to log phase in YPD medium then were transferred to nitrogen starvation medium for 4 h. At different time point, 5 $OD_{600}$ units of cells were collected, washed and resuspended in ice-cold assay buffer (250 mM Tris-HCl, pH 9; 10 mM $MgSO_4$ and 10 µM $ZnSO_4$) with 1 mM PMSF. Then cells were broken using glass beads. For the assay, 10 µl of lysed cells are added to 500 µl of ice-cold assay buffer, placed at 30 °C for 5 min before tadding 50 µl of 55 mM α-naphthyl phosphate disodium salt for 20 min at 30 °C. The reaction was stopped with 500 µl of 2 M glycine-NaOH, pH 11 and the fluorescence measured (345 nm excitation /472 nm emission). The Pho8Δ60 activity corresponds to light emission per amount of protein in the reaction (mg) and reaction time (min).

The number of vacuoles was counted after incubation of exponentially growing cells with FM4-64 (33 µM) in YPD medium at 30 °C for one hour, washing and imaging.

For survival to nitrogen starvation cells were grown to log phase in appropriate complete synthetic medium (CSM) and transferred to nitrogen starvation medium (0.17% yeast nitrogen base and 2% glucose). After 0–6 days of starvation, cells were spread on YPD plates and colonies were counted after 2 days at 30 °C. For LGG-1 rescue assays, LGG-1(G116A) and LGG-1(G116AG117*) were generated by PCR amplification from cDNA LGG-1 and cloned in pRS416 vector under the control of GPD promoter.

