## [Editor Report]

The ubiquitin-like ATG8 family members act at multiple steps of autophagy, such as in autophagosome formation, cargo recognition and autophagosome maturation. ATG8 family members are lipidated that is thought to be required for their function. In this study, the authors provide evidence to show that the *C. elegans* ATG8 homolog LGG-1 possesses lipidation-independent function in autophagy, providing a novel insight into the role of ATG family members during animal development.

---

## [Decision Letter]

[Editors' note: this paper was reviewed by Review Commons.]

Thank you for submitting your article "LGG-1/GABARAP lipidation is not required for autophagy and development in *C. elegans*" for consideration by *eLife*. Your article has been reviewed by 3 peer reviewers at Review Commons, and the evaluation at *eLife* has been overseen by a Reviewing Editor and Benoît Kornmann as the Senior Editor.

Based on the previous reviews and the revisions, the manuscript has been improved but there are some remaining issues that need to be addressed, as outlined below:

In summary, all three reviewers found that you had not addressed the major criticism, namely that the paper has no assays for GABARAP lipidation or membrane association. This claim that GABARAP is not lipidated is central to your study and is reflected in the title. It is therefore crucial that this point be addressed.

*Reviewer 1:*

The rebuttal's Reply #26 argues against doing immuno-EM, but it could be a valuable experiment to do in terms of addressing if there is any LGG-1 protein on the structures shown in Figure 6C-E, K (Type 1 and 2 examples are not shown, making that figure especially difficult to evaluate – structures/arrows should be explained in all cases). Moreover, testing if the 'unknown' band * is modified in atg-4 mutants could be insightful; this band should be commented on more explicitly inside the manuscript, or it will likely prompt questions from readers as it did for all the reviewers.

The manuscript is missing comments about the number of repeats essentially everywhere, and the new Figure S3*, along with other figures, are not quantified. Moreover, error bars are missing in several places and/or are not explained. It is also not clear how many times strains were outcrossed (methods say that the new CRISPR strains were outcrossed, but not how many times – same with lgg-2 mutants).

(*new Figure S3, ie GFP::LGG-1 and GFP::LGG-1(G114A) analyzed following heat shock; the control (no RNAi) experiments were carried out in Kumsta et al., Nat Comm, 2017, but this is not mentioned.)

Below is a summary of additional comments to improve read of text/figures:

1) Line 119 on page 4 – LGG-1 P and LGG-1 I should be defined (aligned with use elsewhere, eg. line 131).

2) Line 423 on page 11 – a conclusion/closing statement is missing.

3) Yeast data has been moved to supplements but the data are now poorly integrated into the text with no conclusion on, line 229.

4) Figure 1 M is out of order, suggest making it new 1Q.

5) Figure 3 and 4I-J should have micrographs in green, to keep with the formatting of the rest of the manuscript's figures.

6) Figure 5K – western blot should have the different LGG-2 isoforms labeled; suggest deleting purple/last lane as these data are included in another blot in supplements, and the rest of Figure 5 does not analyze this strain.

7) Text describing Figure 7 on page 8 is in places incorrect with regards to reference to the figures; authors are encouraged to describe ATG-18 data first, then all their SEPA-1 data.

8) New Figure S7I should be formatted for pair-wise comparisons, as is Figure S7O.

9) Past tense still not corrected in several instances, e.g. p.5, line 192 and p. 7, line 268, and nomenclature should be visited (missing spacing in all genotypes with two genetic loci)

*Reviewer 2:*

The paper has no assays for GABARAP lipidation or membrane association. PE lipids are found on most organelles, including the ER, making it difficult to distinguish non-specific membrane localization from diffuse cytosolic localization using low-magnification images and wide-field illumination. Throughout the manuscript, they refer to "the membrane" when I think they need to specify "the autophagosome membrane".

*Reviewer 3:*

Given the potential impact of the conclusions, the authors would have to provide data that clearly demonstrates a lack of lipidation or membrane association.

---

## [Author Response]

1. General Statements [optional]

First of all, we would like to thank the reviewers for their constructive comments and their suggestions, which were very helpful in planning and performing new experiments and significantly improved our work. In this thoroughly revised version, we have added new experiments. All of the main figures (and four of the supplementary figures) were changed and two new supplementary figures were added. The major changes we made in the revised manuscript are the followings:

1. New data supporting that the LGG-1(G116A) mutant does not localize to the membrane even when autophagy is strongly induced (SupFig4).

2. The electron microscopy of the double mutants lgg-1(G116A) lgg-2(null) and lgg1(G116AG117*) lgg-2(null) (Fig6) showing the partial implication of LGG-2, the LC3 homolog in *C. elegans*.

3. The quantitative analysis of the degradation of a second cargo (protein aggregates) (Fig7) reveals that the lipidated form of LGG-1 allows the coordination between cargo recognition and autophagosome biogenesis.

Below are our specific and detailed responses to the reviewers' comments.

2. Point-by-point description of the revisionsReviewer #1 (Evidence, reproducibility and clarity (Required)):The manuscript by Laboutet et al., titled: "LGG-1/GABARAP lipidation is dispensable for autophagy and development in *C. elegans*," describes the potential function of a nonlipidated LGG-1 mutant containing a G116A mutation. Comparison of a G116A missense mutation to the lgg-1 null mutation or a lgg-1(G116A>G117*) suggests that there is some function retained in the G116A missense mutation. The authors claim that no foci form in the lgg-1(G116A) mutants and take this to mean that there is no lipidation. Assays for autophagy function are carried out, such as the degradation of paternal mitochondria in the 1-cell and 15-cell embryo, survival after L1 starvation, normal lifespan, and the presence of apoptotic corpses. In all cases, the lgg-1(G116A) mutant clearly shows function. However, how can we be sure that there is no lipidated form? The authors state that not seeing LGG-1 positive dots in the embryos with an LGG-1 antibody is enough to state that this is not a lipidated form of LGG-1. However, this should be confirmed biochemically. If there were absolutely no lipidated form, the authors also would have to confirm that the function that they see in their assays, for example in survival after starvation, or in degradation of paternal mitochondria is indeed autophagy-dependent. Double mutants with the lgg-1(G116A) and a degradation mutant, like epg-5, should eliminate the activity seen in their assays. Otherwise, this activity may be due to another function of LGG1 that is not autophagy-dependent.Major questions:1. Can we be sure that there is no lipidated form? What if another amino acid can be lipidated to a lower extent ?

During the revision, we performed several approaches to further demonstrate the absence of lipidation in the G116A mutant.

We attempted to perform a new mass spectrometric analysis of the wild-type and mutant LGG-1 proteins based on the protocol described by the Florey lab (*Durgan et al. Mol Cell, 2021*). This approach has been used to identify and quantify the lipidation of LC3/GABARAP to either a phosphatidyl ethanolamine or a phosphatidyl serine during LC3 associated phagocytosis. The authors purified overexpressed LC3 and GABARAP proteins tagged by GFP and after saponification treatment analyzed them by mass spectrometry.

We followed the published protocol for detecting LGG-1 lipidation in *C. elegans* using an LGG-1 specific antibody for immuno-precipitation. However, we did not detect a C-terminal peptide with the PE moiety. These data suggest that the sensitivity is not sufficient when working on the endogenous protein.

Because lipidation is associated with membrane localization, we developed an alternative strategy based on the localization of GFP::LGG-1(G116A) and GFP::LGG-1(wt) under conditions of strong accumulation of autophagosomes. Strong autophagy flux in the epidermis was induced by a heat stress (as previously reported in *Chen et al. JCB 2021*) and massive accumulation of GFP::LGG-1(wt) positive autophagosomes was achieved after blocking fusion with the lysosome (*epg-5* or *rab-7*). In contrast, GFP::LGG-1(G116A) remains completely diffuse under similar conditions, demonstrating that the G116A mutation completely prevents targeting to the autophagosomes. These experiments are shown in a new supplementary figure (revised FigureS3).

If it is not lipidated, how do the authors propose that this LGG-1 mutant is functioning?

Figure 7 shows several new experiments performed to understand how the LGG1(G116A) mutant functions to degrade cargoes without membrane localization. Our results support a model in which the cleaved form of LGG-1 is sufficient for initiation of autophagosome biogenesis. The function of LGG-1 form I is dependent on the ULK1 complex (the interaction between LGG-1 and UNC-51 was described by *Wu et al. Mol. Cell, 2015*).

The absence of lipidated LGG-1 is partially compensated by LGG-2 proteins, especially for the late steps of autophagy (see response 4 below).

In the G11 6A mutants, and G116AG117* mutant, a new band shows in between the LGG-1 I and LGG-1 II forms, does this band have any activity?

Because LGG-1(G116A), but not LGG-1(G116AG117*), is capable of performing autophagy, we assume that this new minority band has no autophagy-related function but cannot formally rule out a developmental activity. However, since the minority band is not detected in wild-type animals, it could correspond to either a transient intermediate or some other posttranslational modification associated with the mutation.

What if this activity is not autophagy-dependent?

To visualize autophagosomes, we performed electron microscopy (EM) analysis of the double mutants *lgg-1(G116AG117stop); lgg-2(null)* and *lgg-1(G116A); lgg-2(null).* EM indicates that autophagosomes can be formed in *lgg-1(G116A); lgg-2(null*), although less efficiently, and reinforces the conclusion that LGG-1(G116A) is indeed active for autophagy. These results are now shown in Figure 6. See also response 7 for a new experiment showing that LGG-1(G116A) activity is dependent on the autophagy proteins UNC-51 and EPG-2.

LGG-1(G116A) accumulates mainly as a diffuse signal in the cytosol, indicating that it is not degraded by autophagy. The “foci” are rare and very weak in intensity compared to wild-type. Moreover, foci are also detected in other LGG-1 mutants incapable of autophagy, some of which have a major deletion of the protein (Figure 1). We speculate that they may be caused by fixation treatment for immunofluorescence. A new IF experiment in Figure 7 (L,M) shows that there is no co-localization of LGG-1(G116A) with the SEPA-1 aggregates, whereas LGG-1(wt) puncta overlay them. See also response 1 (part 2) for live imaging of GFP::LGG-1(G116A) in the adult stage. In Figure 4J, staining shows HSP-6::GFP (but not LGG-1) to document delayed degradation of paternal mitochondria in *lgg-1(G116A); lgg2(null)* embryos.

The data are now presented as bar charts to facilitate comparisons, and the statistical analysis is shown in revised Figure S5 (previously S4), which shows that there is a small but significant difference.

There is evidence that the efficiency of degradation by autophagy in aggrephagy is modulated by the composition of the aggregates (Zhang et al. 2017). A model has been proposed where PGL-1, PGL-3 and SEPA-1 are mainly degraded via an EPG-2 mediated pathway, however an EPG-2 independent pathway also exists. Which pathway is being used in the LGG-1(G116A) mutant ?

We thank the reviewer for bringing up this point. In the revised version, we quantified the degradation of SEPA-1 aggregates (Figure 7). To this end, we used RNAi against *epg-2* and *unc-51/ulk1* and monitored SEPA-1 degradation in the LGG-1(G116A) mutant or LGG1(wt). The experiment shown in Supplementary Figure S7 demonstrates that LGG1(G1116A) is dependent on EPG-2 and UNC-51.

The manuscript would benefit from some language editing. In page 2, line 5, it reads: "The general scheme is successive recruitment of a series of protein complexes involved in the dynamic of the process through several steps implicating the phosphorylation of lipids…" Here, it should read "dynamics." The authors use this term often and they should refer to "dynamics".

This has been corrected.

The label "1-cell" are missing in Figure 1B showing the lgg-1() mutant on the left.

The missing "1 cell" label has been corrected (revised Figure 3B).

Reviewer #2 (Evidence, reproducibility and clarity (Required)):Leboutet et al. use a clever strategy to test the role of LC3 modifications in animal cells. They generate an allelic series of cleavage site mutants of the major LC3 isoform in *C. elegans*, LGG1. They convincingly demonstrate that a non-cleavable precursor form of LC3(AA) is unable to localize or function during various forms of macroautophagy, embryonic development, adult survival, or cell death/corpse clearance. A pre-cleaved intermediate form of LC3(A*) is also unable to localize or function during various forms of macroautophagy and has neomorphic characteristics visualized by EM and corpse clearance, but fully functions to promote embryonic development. Surprisingly, mutating the predicted cleavage site of LC3(AG) results in defects in localization, but only a mild delay in autophagic flux. Similarly, LC3(AG) mutants show no defects in viability or embryonic development, which the authors show is partially due to the function of the other LC3 isoform, LGG-2.Major comments:What is the new LGG form * in Figure 1C? Does the Mass Spec data give any hints? The authors imply that this is not lipidated, but show no direct evidence for this statement. There are reports of LC3 conjugation to lipids beside PE, such as PS. Could this represent a switch form LC3-PE to LC3-PS? Or simply cleavage and lipidation at G117? The lack of localization to autophagosomes convincingly demonstrates that this form * does not act like the classic form II, which was thought to be the functional form of LC3, but more information about this isoform would be needed to convincingly make the author's conclusions about lipidation.

See Reply 1 that describes our attempts to characterize the lipidation using mass spectrometry analyses and the new series of experiments using a GFP::LGG-1(G116A) reporter (Supplementary figure S3). The switch from a PE to a PS conjugation described by Durgan and colleagues (*Mol Cell, 2021*) is still associated to the membrane localization of LC3 and GABARAP in mammals. This is not what we observed for LGG-1(G116A) supporting absence of conjugation to either PE or PS. The mass spectrometry analyses shown in supplementary figure S2 have been repeated several times and identified the precursor and the cleaved form after A116 but not after the G117.

The text compares the number of omegasomes vs phagophores vs autophagosomes and refers to Figure 7E-G, but these graphs do not clearly identify the number of double-positive and singlepositive populations, making it impossible to interpret this data. A graph similar to Figure S5A should replace 7E-G to clearly convey this data.

The graph has been corrected as requested by the reviewer (revised Figure 7G).

Figure 7E vs 7P – Why are there twice as many ATG-18 dots in 7P controls? Is one OP50-fed and the other HT115-fed? Or are the strains different? Why this is different isn't clear from the methods and is missing from the worm strain list.

The experiments shown in Figures 7E and 7P are independent experiments with HT115-fed animals, each performed at least three times. However, one experiment involves immunofluorescence with two antibodies (GFP and LGG-2) whereas the other one involves live imaging of the GFP signal. The background noise in IF and the better contrast in live imaging could explain the differences. In both cases, the differences between the mutants are similar. The method section was corrected to better explain the live imaging, and supplementary table 1 lists the atg-18::GFP strains.

Figure S4F – I'm not sure of the utility of the LGG-1 rescue experiments in yeast. WT LGG-1 expression doesn't appear to significantly rescue atg8∆ mutants and it's not clear that there is any significant difference between different LGG-1 isoforms, especially given the broken y-axis. Also showing n=1 and missing statistics. The other yeast experiments are more interpretable and these findings do not significantly add to the paper.

The data are now presented as bar charts, and the statistical analysis can be seen in the revised Figure S5 (previously S4), which shows a small but significant difference between LGG-1(G116A). In the revised version, all yeast experiments have been moved to the supplementary data with the “Results” and “Material and Methods” sections.

Minor comments:First half of the first paragraph of the introduction is under-referenced. Please cite relevant review articles. Introduction could also be shortened and more to the point.

The introduction was shortened by 30% and more recent reviews were added in the first paragraph.

Missing statistics in Figure 1L right. Can't conclude it's increased if not significant.

We thank the reviewer for pointing out this error. The absence of an asterisk (p value <0.05) from the previous version of Figure 1 has now been corrected.

Figure 1N is not discussed in the manuscript.

Figure 1N is a control showing that altering cleavage of the LGG-1 precursor by ATG-4.1 depletion reduces but does not abolish subsequent lipidation (*Wu et al., J Biol Chem. 2012*), as observed by fewer and weaker puncta. Such puncta are not visible in LGG1(G116A), providing an indirect argument for the absence of lipidation in this mutant. The text has been changed accordingly.

Figure 3 would be improved by maintaining the color scheme from Figure 2

A similar color code is now present in the two figures (Figure 3 is now the revised Figure 2).

Figure 3H and Figure 4D are showing similar data in opposite ways (viability vs. lethality). For your reader's sake, please use the same measure for the same assay.

The plot was homogenized in the revised Figure 4D.

There is no 5-cell stage. *C. elegans* early embryonic stages are 1, 2, 3, 4, 6, 7, 8, 12, 14, 15.

This error has now been corrected in the revised Figure 6B.

The relative prevalence of LGG-2-I vs LGG-2-II should be presented in Figure 5K, similar to the analysis of LGG-1 isoforms in Figure 1C. It appears that LGG-2 conjugation is being altered in various lgg-1 alleles.

Quantification for the two bands is now shown in the revised Figure 5.

Figure 6H – EM counts are typically represented as number per section area, not section. The size of cell sections can vary by a large amount.

The reviewer is correct, and the graph in the revised Figure 6 has been corrected to indicate quantification per cut surface.

The authors refer to G116AG117* as gain-of-function, but this is confusing given all the LGG-1 functions lost. A more accurate term could be neomorphic, although the authors haven't performed the genetics to test whether the allele is antimorphic (i.e. G116AG117*/null ).

We thank the reviewer for the constructive comment. EM analysis with double mutants supports the use of neomorphic (see response 1 and revised Figure 6) and has been corrected in the text.

Why wasn't the double alanine mutant used in any assays past Figure 3?

The LGG-1(G116AG117A) mutant does not allow autophagy and has a strong developmental phenotype with greatly reduced viability (see revised Figure 2). Genetic crosses and other experiments are more complicated to achieve. However, in the new experiments shown in the revised Figure 7, the *lgg-1(G116AG117A)* mutant was analyzed.

Figure 7R right model – Phagophore membranes need to be connected at the ends – What are the light green circles representing?– Why does the blue G116A mutant localize to the cargo in the model? The author's said they didn't observe any localization.

We recognized that the model was more confusing than helpful and decided to remove it from the revised version.

Why is Figure 2N identical to Figure S3D? There's no need to include the same data twice. Also, both contain an error on the y-axis (15 instead of 5).

The typo on the y-axis has been corrected and Figure S3D has been removed.

c – "Our genetic data indicate that form I of LGG-1 is sufficient for initiation, elongation and closure of autophagosomes". Indicate is an overstatement. The authors do not perform assays for initiation, elongation or closure.

EM and marker analyses show that autophagosomes are formed, but we did not precisely quantify each step. Thus, we have changed "indicate" to "suggest".

Discussion – P. 12 – "paternal mitochondria could be degraded by autophagosomes devoid of both LGG-1 and LGG-2 " – I couldn't find data in this paper where paternal mitochondria are shown to never have LGG-1 or LGG-2 on them. A single time point analysis isn't sufficient to demonstrate that for molecules that dynamically associate and disassociate with membranes.

This hypothesis is based on the double mutant lgg-1(G116A);lgg-2(null). In this strain, there is no LGG-2 protein and LGG-1(G116A) does not form puncta. We have now added an EM analysis showing that the paternal mitochondria in this strain are sequestered in autophagosomes (revised Figure 6). The sentence in the text begins with “It suggests…”

*To this end, an unaddressed concern in this study is that it has not been ruled out if LGG1(G116A) perhaps can still trigger an unspecified entity to associate with membranes.*
Specifically, the authors identify a lower band (referred to as an unexpected, minor band) in Figure 1C for G116A and G116AG117A, but do not investigate the nature of this band (noting, importantly, that these two mutants show normal development ). Immuno-EM could be very useful here.

Our results demonstrate that the mutant LGG-1 proteins are not addressed to membranes. ImmunoEM could be performed, but it is a lengthy experiment in which we expect a negative result for membrane localization. We proposed alternative strategies to increase the number of autophagosomes and further exclude weak localization to membranes. See Reply 1.

Several LGG-1 mutants exhibit a drastic developmental phenotype and greatly reduced viability (see revised Figure 2). Breeding and synchronization of these strains is difficult and, therefore, has been limited to experiments essential for the purpose of this work.

We performed EM analysis of the double mutants, which is now shown in the revised Figure 6. We can still observe autophagosomes in the *lgg-1(G116A); lgg-2(null)* mutant, but almost none in the *lgg-1(G116A G116STOP); lgg-2(null)* mutant. We thank the reviewer for his/her constructive suggestion on Atg-4, but feel that this is outside the scope of this manuscript. We did not detect heterodimerization of LGG-1 and LGG-2 by mass spectrometric analyses after immunoprecipitation (our unpublished data) but cannot exclude this possibility.

The reviewer points out several important experiments that were performed for the revision. In particular, the EM of double mutants is shown in the revised Figure 6 (see response 28). We performed a series of experiment with GFP::LGG-1(wt) and GFP::LGG1(G116A) reporters. They confirmed that the G116A mutation completely abolishes the localization of LGG-1 to autophagosomes even in the presence of strong autophagy induction. GFP::LGG-1(G116A) is well suited as a negative control to detect the absence of GFP aggregation due to overexpression (see the new Supplementary Figure S3).

Figure 2N and S2D are replicated.

Panel D from Supplementary Figure S3 has been removed.

This has been corrected.

LGG-2 bands on Western blot have been quantified.

Figure 7 feels like almost 'walking' backwards, may be more efficiently integrated elsewhere in the manuscript (it is also not clear why lgg-2 RNAi is used here, instead of the mutants that are used everywhere else in the study?). Moreover, the authors may want to consider discussing Figure 3/development first (considering the reader has been informed that lgg-1 is an essential gene,- to this point, it is only later made clear that the lethal allele has 8% 'breakthroughs – are these the animals analyzed?) and Figure 6/EM together with Figure 1.

We thank the reviewer for his/her constructive suggestions. The outline of the results has been changed in the revised version, now presenting the developmental studies earlier (revised Figure 2). The escapers of the lethal alleles of *lgg-1* are those used for the experiments. Because breeding of these strains and genetic crosses are difficult, we used RNAi approaches for several experiments. However, we did not use RNAi against lgg-2.

The yeast section is highlighted in the abstract whereas all data are in supplements; overall it could be better integrated. In particular, sequence alignments and Western blots are missing here.

The alignment of ScAtg8 is now shown in the revised Figure S1. We tried using antibodies to perform Western blots, but the commercial antibodies do not work well for yeast Atg8. As discussed in response 11, we decided to move the yeast data to the supplemental results and removed the sentence from the abstract.

Result section should be revisited for clarity and language, including written in past tense.

The result section has been edited and checked for the correct use of tenses.

References

Durgan, J., Lystad, A. H., Sloan, K., Carlsson, S. R., Wilson, M. I., Marcassa, E., Ulferts, R., Webster, J., Lopez-Clavijo, A. F., Wakelam, M. J., Beale, R., Simonsen, A., Oxley, D., & Florey, O. (2021). Non-canonical autophagy drives alternative ATG8 conjugation to phosphatidylserine. Molecular cell, 81(9), 2031–2040.e8. https://doi-org.insb.bib.cnrs.fr/10.1016/j.molcel.2021.03.020

Wu, F., Li, Y., Wang, F., Noda, N.N., and Zhang, H. (2012). Differential Function of the Two Atg4 Homologues in the Aggrephagy Pathway in *Caenorhabditis elegans*. J Biol Chem *287*, 29457–29467. https://doi.org/10.1074/jbc.M112.365676.

Wu, F., Watanabe, Y., Guo, X.-Y., Qi, X., Wang, P., Zhao, H.-Y., Wang, Z., Fujioka, Y., Zhang, H., Ren, J.-Q., et al. (2015). Structural Basis of the Differential Function of the Two *C. elegans* Atg8 Homologs, LGG-1 and LGG-2, in Autophagy. Molecular Cell *60*, 914–929. https://doi.org/10.1016/j.molcel.2015.11.019.

[Editors' note: further revisions were suggested prior to acceptance, as described below.]

Reviewer 1:The rebuttal's Reply #26 argues against doing immuno-EM, but it could be a valuable experiment to do in terms of addressing if there is any LGG-1 protein on the structures shown in Figure 6C-E, K (Type 1 and 2 examples are not shown, making that figure especially difficult to evaluate – structures/arrows should be explained in all cases). Moreover, testing if the 'unknown' band * is modified in atg-4 mutants could be insightful; this band should be commented on more explicitly inside the manuscript, or it will likely prompt questions from readers as it did for all the reviewers.

Immuno-EM was performed using strains expressing GFP::LGG-1 and GFP::LGG1(G116A) as previously described in Manil-Ségalen et al. 2014 “Antibodies against LGG1 and LGG-2 did not show a sufficient EM signal to quantify the staining of endogenous proteins in embryos.” This new experiment indicates that most of the autophagosomes were not labelled by GFP::LGG-1(G116A) and that when gold beads were detected they were in majority inside the lumen. The results are illustrated in the supplementary Figure S3 (now panels E and F of Figure 1—figure supplement 3) and in the text (page 5 lines 178182).

The legend of Figure 6 has been corrected to mention the type for each panel and explain the white arrow in panel H (type II). Specifically, type1 autophagosomes are shown in (A, C, D), type 2 autophagosomes are shown in (E, white arrow in H, J) and type 3 vesicles are shown in (G, I).

Here, we focused our efforts on immunoEM and fractionation experiments and did not further explore the *atg-4* mutants. We have briefly discussed the nature of the unknown minority band, which is probably a non-functional form only present in LGG1(G116AG117*) and LGG-1(G116AG117A) (page 10 lines 397-400).

The manuscript is missing comments about the number of repeats essentially everywhere, and the new Figure S3*, along with other figures, are not quantified. Moreover, error bars are missing in several places and/or are not explained. It is also not clear how many times strains were outcrossed (methods say that the new CRISPR strains were outcrossed, but not how many times – same with lgg-2 mutants).

Supplementary Figure S3 has been quantified (panel D in now called *Figure 1—figure supplement 3*). Error bars are standard deviations, which is now indicated in the material and methods section (page 20 line 606).

We also added that the experiments were done at least three times (page 20 line 600). The number of outcrosses has been indicated.

Missing error bars have been corrected in Fig3F, Fig4D, FigS4B.EM experiments have no error bars because the experiments are not independent, due to technical constraints of the cryo-fixation and cryo-substitution. For each strain analyzed, a series of blocs containing multiple animals have been processed together. The numbers of animal and sections observed are important (as indicated in the legend of the figure) because the frequencies of autophagosomes can be low and the data from several experiments have been pooled, which is now indicated in the figure legend.

(*new Figure S3, ie GFP::LGG-1 and GFP::LGG-1(G114A) analyzed following heat shock; the control (no RNAi) experiments were carried out in Kumsta et al., Nat Comm, 2017, but this is not mentioned.)

We apologize for this flaw. The reference has been added.

Below is a summary of additional comments to improve read of text/figures:1) Line 119 on page 4 – LGG-1 P and LGG-1 I should be defined (aligned with use elsewhere, eg. line 131).

LGG-1 P and LGG-1 I are now defined at the end of the introduction (page 3 line 102).

2) Line 423 on page 11 – a conclusion/closing statement is missing.

We added a conclusive sentence (page 11 line 436).

3) Yeast data has been moved to supplements but the data are now poorly integrated into the text with no conclusion on, line 229.

A conclusion has been added to the paragraph (page 6 line 239).

4) Figure 1 M is out of order, suggest making it new 1Q.

Figure 1 has been revised and previous panels; M-Q have been moved to supplementary Figure S2 (now *Figure 1—figure supplement 2*).

5) Figure 3 and 4I-J should have micrographs in green, to keep with the formatting of the rest of the manuscript's figures.

Single color images are generally shown in grey levels because the contrast is better compared to green (Figures 1, 3, 4, 5, S3, S4, and S5) with the exceptions of Figures 7AE and S7 A-H.

6) Figure 5K – western blot should have the different LGG-2 isoforms labeled; suggest deleting purple/last lane as these data are included in another blot in supplements, and the rest of Figure 5 does not analyze this strain.

Figure 5K has been modified as requested.

7) Text describing Figure 7 on page 8 is in places incorrect with regards to reference to the figures; authors are encouraged to describe ATG-18 data first, then all their SEPA-1 data.

Text (page 8 from line 323) and Figure 7 have been modified to describe ATG-18 data first, then SEPA-1 data. References to the figures have been corrected.

8) New Figure S7I should be formatted for pair-wise comparisons, as is Figure S7O.

Supplementary Figure S7 (now *Figure 5—figure supplement 1*) has been formatted for pairwise comparison.

9) Past tense still not corrected in several instances, e.g. p.5, line 192 and p. 7, line 268, and nomenclature should be visited (missing spacing in all genotypes with two genetic loci)

We thank the reviewer for pointing these mistakes, which have been corrected in the revised manuscript.

Reviewer 2:The paper has no assays for GABARAP lipidation or membrane association. PE lipids are found on most organelles, including the ER, making it difficult to distinguish non-specific membrane localization from diffuse cytosolic localization using low-magnification images and wide-field illumination. Throughout the manuscript, they refer to "the membrane" when I think they need to specify "the autophagosome membrane".

A new and completely independent biochemical approach has been performed to fractionate membrane-bound proteins. The results clearly demonstrated that the LGG1(G116A) and the LGG-1(G116AG117*) are not detectable in membrane fractions containing autophagosomes, contrary to the wild-type LGG-1 and LGG-2. This novel and important result is described in the revised Figure 1 (panel M) and in the result section (page 5 lines 166-70). A new sub-section describing the protocol has been added to the material and methods section. These data strongly support the key finding that the addressing of LGG-1 to the membrane is dispensable for the autophagy and developmental functions.

We have checked the manuscript, and we have now specified “the autophagosome membrane” when the text was specifically related with autophagy process. Because LC3/GABARAP proteins can also be conjugated to other membranes, in few places when the text was not specifically referring to autophagy, we kept “the membrane”.

Reviewer 3:Given the potential impact of the conclusions, the authors would have to provide data that clearly demonstrates a lack of lipidation or membrane association.

See our response to reviewer #2 and the new figure panel 1M, which shows cellular membrane fractionation to monitor membrane association of wild-type and mutant LGG-1 proteins by an independent biochemical approach.